# PatchGT: Transformer over Non-trainable Clusters for Learning Graph Representations

**Han Gao***
University of Notre Dame
hgao1@nd.edu

**Xu Han***
Tufts University
Xu.Han@tufts.edu

**Jiaoyang Huang***
University of Pennsylvania
huangjy@wharton.upenn.edu

**Jian-Xun Wang**
University of Notre Dame
jwang33@nd.edu

**Li-Ping Liu**
Tufts University
Liping.Liu@tufts.edu

## Abstract

Recently the Transformer structure has shown good performances in graph learning tasks. However, these Transformer models directly work on graph nodes and may have difficulties learning high-level information. Inspired by the vision transformer, which applies to image patches, we propose a new Transformer-based graph neural network: Patch Graph Transformer (PatchGT). Unlike previous transformer-based models for learning graph representations, PatchGT learns from non-trainable *graph patches*, not from nodes directly. It can help save computation and improve the model performance. The key idea is to segment a graph into patches based on spectral clustering without any trainable parameters, with which the model can first use GNN layers to learn patch-level representations and then use Transformer to obtain graph-level representations. The architecture leverages the spectral information of graphs and combines the strengths of GNNs and Transformers. Further, we show the limitations of previous hierarchical trainable clusters theoretically and empirically. We also prove the proposed non-trainable spectral clustering method is permutation invariant and can help address the information bottlenecks in the graph. PatchGT achieves higher expressiveness than 1-WL-type GNNs, and the empirical study shows that PatchGT achieves competitive performances on benchmark datasets and provides interpretability to its predictions. The implementation of our algorithm is released at our Github repo: https://github.com/tufts-ml/PatchGT.

## 1 Introduction

Learning from graph data is ubiquitous in applications such as drug design [15] and social network analysis [37]. The success of a graph learning task hinges on effective extraction of information from graph structures, which often contain combinatorial structures and are highly complex. Early works [7] often need to manually extract features from graphs before applying learning models. In the era of deep learning, Graph Neural Networks (GNNs) [35] are developed to automatically extract information from graphs. Through passing learnable messages between nodes, they are able to encode graph information into vector representations of graph nodes. GNNs have become the standard tool for learning tasks on graph data.

While they have achieved good performances in a wide range of tasks, GNNs still have a few limitations. For example, GNNs [36] suffer from issues such as inadequate expressiveness [36], over-smoothing [28], and over-squashing [2]. These issues have been partially addressed by techniques such as improving message-passing functions and expanding node features [5, 21].

---

*Equal contribution.

H. Gao et al., PatchGT: Transformer over Non-trainable Clusters for Learning Graph Representations. *Proceedings of the First Learning on Graphs Conference (LoG 2022)*, PMLR 198, Virtual Event, December 9–12, 2022.

Another important progress is to replace the message-passing network with the Transformer architecture [6, 18, 24, 38]. These models treat graph nodes as tokens and apply the Transformer architecture to nodes directly. The main focus of these models is how to encode node information and how to incorporate adjacency matrices into network calculations. Without the message-passing structure, these models may overcome some associated issues and have shown premium performances in various graph learning tasks. However, these models suffer from computation complexity because of the global attention on all nodes. It is hard to capture the topological information of graphs.

As a comparison, the Transformer for image data works on image patches instead of pixels [9, 22]. While this model choice is justified by reduction of computation cost, recent work [31] shows that "patch representation itself may be a critical component to the 'superior' performance of newer architectures like Vision Transformers". One intriguing question is whether patch representation can also improve learning models on graphs. With this question, we consider patches on graphs. Patches over graphs are justified by a "mid-level" understanding of graphs: for example, a molecule graph's property is often decided by some *function groups*, each of which is a subgraph formed by locally-connected atoms. Therefore, patch representations are able to capture such mid-level concepts and bridge the gap between low-level structures to high-level semantics.

Motivated by our question, we propose a new framework, Patch Graph Transformer (PatchGT). It first segments a graph into patches based on spectral clustering, which is a non-trainable segmentation method, then applies GNN layers to learn patch representations, and finally uses Transformer layers to learn a graph-level representation from patch representations. This framework combines the strengths of two types of learning architectures: GNN layers can extract information with message passing, while Transformer layers can aggregate information using the attention mechanism. To our best knowledge, we firstly show several limitations of previous trainable clustering method based on GNN. We also show that the proposed non-trainable clustering can provide more reasonable patches and help overcoming information bottleneck in graphs.

We justify our model architecture with theoretical analysis. We show that our patch structure derived from spectral clustering is superior to patch structures learned by GNNs [4, 13, 39]. We also propose a new mathematical description of the information bottleneck in vanilla GNNs and further show that our architecture has the ability of mitigating this issue when graphs have small graph cuts. The contributions of this paper are summarized as follows.

- We develop a general framework to overcome the information bottleneck in traditional GNNs by applying a Transformer on graph patches in Section 3. The graph patches are from an unlearnable spectral clustering process.
- We prove several new theorems for the limitations of previous pooling methods from the 1-WL algorithm in Theorem 1 and Theorem 2. And we theoretically prove that PatchGT is strictly beyond 1-WL and hence has better expressiveness in Theorem 3. Also, in Section 4.4, we show that the segmentations from hierarchical learnable clustering methods may aggregate disconnected nodes, which will definitely hurt the performance of the transformer model.
- We demonstrate the existence of information bottleneck in GNNs in Section 4.3. When a graph consists of loosely-connected clusters, we make the first attempt to characterize such information bottleneck. And it indicates when there is a small graph cut between two clusters, the GNNs need to use more layers to pass signals from one group to another. And we further demonstrate with direct attention between groups, PatchGT could overcome such limitations.

We run an extensive empirical study and demonstrate that the proposed model outperforms competing methods on a list of graph learning tasks. The ablation study shows that our PatchGT is able to combine the strengths of GNN layers and Transformer layers. The attention weights in Transformer layers also provide explanations for model predictions.

## 2 Related Work

Transformer models have gained remarkable successes in NLP applications [16]. Recently, they have also been introduced to vision tasks [9] and graph tasks [6, 11, 18, 20, 24, 34, 38, 40]. These models all treat nodes as tokens. Particularly, Memory-based graph networks[1] apply a hierarchical attention pooling methods on the nodes. GraphTrans [34] directly applies a GNN on all nodes, followed by a transformer. Therefore, they are hard to be applied to large graphs because of huge computation complexity.

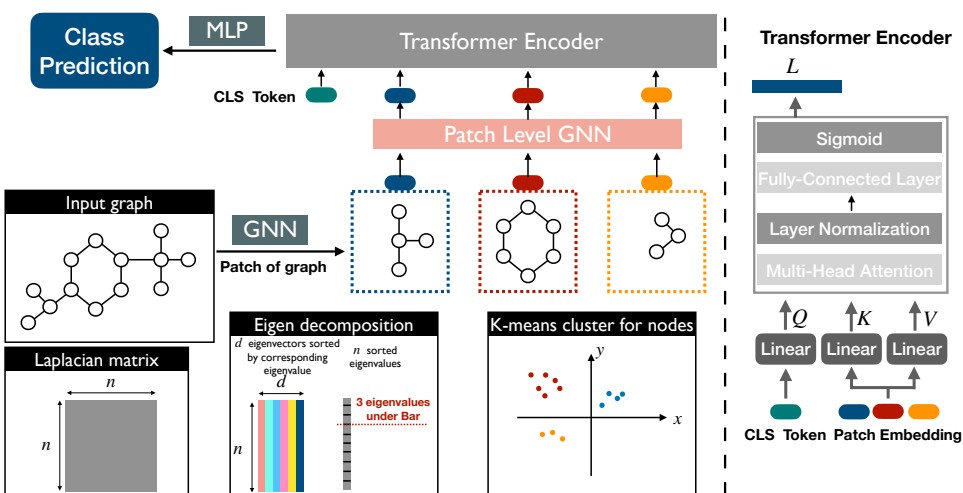

**Figure 1:** Model review. We segment a graph into several patch subgraphs by non-trainable clustering. We first extract local information through a GNN, and the initial patch representations are summarized by the aggregation of nodes within the corresponding patches. To further encode structure information, we apply another patch-level GNN to update the representations of patches. Finally, we use Transformer to extract the representation of the entire graph based on patch representations.

At the same time, image patches have been shown to be useful for Transformer models on image data [9, 31], so it is not surprising if graph patches are also helpful to Transformer models on graph data. Graph multiset pooling [3] applies trainable pooling methods on the nodes based on GNN. And then adopt a global attention layer on learned clusters. We will show that such trainable clustering has several limitations for attention mechanism in this work.

Hierarchical pooling models [4, 12, 13, 19, 27, 39] are relevant to our work in that they also aggregate information from node representations in middle layers of networks. However, these methods all form their pooling structures based on representations learned from GNNs. As a result, these pooling structures inherit drawbacks from GNNs [36]. They may also aggregate nodes that are far apart on the graph and thus cannot preserve the global structure of the input graph. Also such trainable clustering methods need much computation for training. Furthermore, our main purpose is to use non-trainable patches on graphs as tokens for a Transformer model, which is different from these models.

## 3 Patch Graph Transformer

### 3.1 Background

In this work, we consider graph-level learning problems. Let $G = (V, E)$ denote a graph with node set $V$ and edge set $E$. Let $\mathbf{A}$ denote its adjacency matrix. The graph has both node features $\mathbf{X} = (\mathbf{x}_i \in \mathbb{R}^d : i \in V)$ and edge features $\mathbf{E} = (\mathbf{e}_{i,j} \in \mathbb{R}^{d'} : (i, j) \in E)$. Let $y$ denote the label of graph. This work aims to learn a model that maps $(\mathbf{A}, \mathbf{X}, \mathbf{E})$ to a vector representation $\mathbf{g}$, which is then used to predict the graph label $y$.

**GNN layers.** A GNN uses node vectors to represent structural information of the graph. It consists of multiple GNN layers. Each GNN layer passes learnable messages and updates node vectors. Suppose $\mathbf{H} = (\mathbf{h}_i \in \mathbb{R}^{d''} : i \in V)$ are node vectors, a typical GNN layer updates $\mathbf{H}$ as follows.

$$\mathbf{h}_i' = \sigma(\mathbf{W}_1\mathbf{h}_i + \sum_{j:(i,j)\in E} \mathbf{W}_2\mathbf{h}_j + \mathbf{W}_3\mathbf{e}_{i,j}) \tag{1}$$

Here matrices $(\mathbf{W}_1, \mathbf{W}_2, \mathbf{W}_3)$ are all learnable parameters; and $\sigma$ is the activation function. We denote the layer function by $\mathbf{H}' = \text{GNN}(\mathbf{A}, \mathbf{E}, \mathbf{H})$. If there are no edge features, then the calculation can be written in matrix form.

$$\mathbf{H}' = \sigma(\mathbf{H}\mathbf{W}_1^\top + \mathbf{A}\mathbf{H}\mathbf{W}_2^\top) \tag{2}$$

## 3.2 Model design

PatchGT has three components: segmenting the input graph into patches, learning patch representations, and aggregating patch representations into a single graph vector. The overall architecture is shown in Figure 1. The second and third steps are in an end-to-end learning model. Graph segmentation is outside of the learning model, which will be justified by our theoretical analysis later.

**Forming patches over the graph.** We first discuss how to form patches on a graph. One consideration is to include an informative subgraph (e.g., a function group, a motif) into a single patch instead of segmenting it into pieces. A reasonable approach is to run node clustering on the input graph and treat each cluster as a graph patch. If a meaningful subgraph is densely connected, it has a good chance of being contained in a single cluster.

In this work, we consider spectral clustering [30, 41] for graph segmentation. Let $\mathbf{L} = \mathbf{I} - \mathbf{D}^{-1/2}\mathbf{A}\mathbf{D}^{-1/2}$ be the normalized Laplacian matrix of $G$, and its eigen-decomposition is $\mathbf{L} = \mathbf{U}\boldsymbol{\Lambda}\mathbf{U}^\top$, where the eigen-values $\boldsymbol{\Lambda} = \mathrm{diag}(\lambda_1, \ldots, \lambda_{|V|})$ is sorted in the ascending order. By thresholding eigen-values with a small threshold $\gamma$, we get $k = \arg\max_{k'} \lambda_{k'} \leq \gamma$ eigen-vectors $\mathbf{U}_{1:k}$, then we run $k$-means to get $k$ clusters (denoted by $\mathcal{P}$) of graph nodes. Here $\mathcal{P} = \{C_{k'} \subset V : k' = 1, \ldots, k\}$ with each $C_{k'}$ representing a cluster/patch. Note that the threshold $\gamma$ is a hyper-parameter, and $k$ varies depending on the underlying graph's topology.

**Computing patch representations**. When we learn representations of patches in $\mathcal{P}$, we consider both node connections within the patch and also connections between patches. Patches form a coarse graph, which is also referred as a patch-level graph, by treating patches as nodes and their connections as edges. We first learn node representations using GNN layers. Let $\mathbf{H}_0 = \mathbf{X}$ denote the initial representations of all nodes. Then we apply $L_1$ GNN layers to get node representations $\mathbf{H}_{L_1}$.

$$\mathbf{H}_\ell = \mathrm{GNN}(\mathbf{A}, \mathbf{E}, \mathbf{H}_{\ell-1}), \ell = 1, \ldots, L_1 \tag{3}$$

Here for easier discussion, we apply GNN layers to the entire graph. We have also tried to apply GNN layers within each patch only and found that the performance is similar.

Then we read out the initial patch representation by summarizing representations of nodes within this patch. Let $\mathbf{z}_{k'}^0$ denote the initial patch representation, then

$$\mathbf{z}_{k'}^0 = \frac{|C_{k'}|}{|V|} \cdot \mathrm{readout}(\mathbf{h}_i^{L_1} : i \in C_{k'}), k' = 1, \ldots, k \tag{4}$$

Here $\mathbf{h}_i^{L_1}$ is node $i$'s representation in $\mathbf{H}_{L_1}$. We collectively denote these patch representations in a matrix $\mathbf{Z}_0 = (\mathbf{z}_{k'}^0 : k' = 1, \ldots, k)$. The readout function $\mathrm{readout}(\cdot)$ is a function aggregating information from a set of vectors. Our implementation uses the max pooling. We use the factor $\frac{|C_{k'}|}{|V|}$ to assign proper weights to patch representations.

To further refine patch representations and encode structural information of the entire graph, we apply further GNN layers to the patch-level formed by patches. We first compute the adjacency matrix $\tilde{\mathbf{A}}$ of the patch-level graph. If we convert the partition $\mathcal{P}$ to an assignment matrix $\mathbf{S} = (S_{i,k'} : i \in V, k' = 1, \ldots k)$ such that $S_{i,k'} = 1[i \in C_{k'}]$, then the adjacency matrix over patches is

$$\tilde{\mathbf{A}} = 1\big[(\mathbf{S}^\top\mathbf{A}\mathbf{S}) > 0\big]. \tag{5}$$

Note that $\tilde{\mathbf{A}}$ only has connections between patches and does not maintain connection strength.

We then compute use $L_2$ GNN layers to refine patch representations.

$$\mathbf{Z}_\ell = \mathrm{GNN}(\tilde{\mathbf{A}}, \mathbf{0}, \mathbf{Z}_{\ell-1}), \quad \ell = 1, \ldots, L_2 \tag{6}$$

GNN layers here do not have edge features. From the last layer, we get patch representations in $\mathbf{Z}_{L_2}$

**Graph representation via Transformer layers.** Then we use $L_3$ Transformer layers to extract the representation of the entire graph. Here we use a learnable query vector $\mathbf{q}_0$ to "retrieve" the global representation $\mathbf{g}$ of the graph from patch representations $\mathbf{Z}_{L_2}$.

$$\mathbf{q}_\ell' = \mathrm{MHA}\left(\mathbf{q}_{\ell-1}, \mathbf{Z}_{L_2}, \mathbf{Z}_{L_2}\right), \quad \ell = 1, \ldots, L_3 \tag{7}$$

$$\mathbf{q}_\ell = \mathrm{MLP}(\mathbf{q}_\ell') + \mathbf{q}_{\ell-1}, \quad \ell = 1, \ldots, L_3 \tag{8}$$

$$\mathbf{g} = \mathrm{LN}(\mathbf{q}_{L_3}) \tag{9}$$

Here $\mathrm{MHA}(\cdot, \cdot, \cdot)$ is the function of a multi-head attention layer (please refer to Chp. 10 of [42]). Its three arguments are the query, key, and value. The two functions $\mathrm{MLP}(\cdot)$ and $\mathrm{LN}(\cdot)$ are respectively a multi-layer perceptron and a linear layer. Note that patch representations $\mathbf{Z}_{L_2}$ are carried through without being updated. Only the query token is updated to query information from patch representations. The final learned graph representation is $\mathbf{g}$, from which we can perform various graph level tasks.

## 4 Theoretical Analysis

In this section, we study the theoretical properties of the proposed model. To save space, we put all proofs in the appendix.

### 4.1 Enhancing model expressiveness with patches

On purpose we form graph patches using a clustering method that is not part of the neural network. An alternative consideration is to learn such cluster assignments with GNNs (e.g. DiffPool [39] and MinCutPool[4]. However, cluster assignment learned by GNNs inherits the limitation of GNNs and hinders the expessiveness of the entire model.

**Theorem 1.** *Suppose two graphs receive the same coloring by 1-WL algorithm, then DiffPool will compute the same vector representation for them.*

Although DiffPool and MinCutPool claims to cluster "similar" graph nodes into clusters during pooling, but these nodes may not be connected. Because of the limitation of GNNs, they may aggregate nodes that are far apart in the graph. For example, nodes in the same orbit always get the same color by the 1-WL algorithm and also the same representations from a GNN, then these nodes always have the same cluster assignment. Merging these nodes into the same cluster does not seem capture the high-level structure of a graph.

Another prominent pooling method is the Graph U-Net [12], which has similar issues. We briefly introduce its calculation here. Suppose the layer input is $(\mathbf{A}, \mathbf{H})$, the model's pooling layer projects $\mathbf{H}$ with a unit vector $\mathbf{p}$ and gets values $\mathbf{v} = \mathbf{H}\mathbf{p}$ for all nodes, then it chooses the top $k$ nodes that have largest values in $\mathbf{v}$ and keep their representations only. We will show that this approach is NOT invariant to node orders.

We also consider a small variant of Graph U-Net for analysis convenience. Instead of choosing $k$ nodes with top values in $\mathbf{v}$, the variant uses a threshold $\beta$ (either learnable or a hyper-parameter) to choose nodes: $\mathbf{b} = \mathbf{v} \geq \beta$. Then the output of the layer is $(\mathbf{A}[\mathbf{b}, \mathbf{b}], \mathbf{H}[\mathbf{b}])$. We call the model with the variant with thresholding as Graph U-Net-th. We show that the variant of Graph U-Net-th is also bounded by the 1-WL algorithm.

**Theorem 2.** *Suppose two graphs receive the same coloring by 1-WL algorithm, then Graph U-Net-th will compute the same vector representation for them.*

The two theorems strongly indicate that pooling structures learned by GNNs have the same drawback. We provide detailed analysis for Graph U-Net in Appendix A.3.

In contrast, a small variant of PatchGT is more expressive than the 1-WL algorithm. Figure 7 in Appendix shows two graph pairs that can be distinguished by PatchGT but not the 1-WL algorithm. In this PatchGT variant, we only need to choose the summation operation to aggregate node representations in the same patch and multiply a scalar to the MHA output. We put the result in the following Theorem.

**Theorem 3.** *Suppose a PatchGT uses GIN layers, uses sum-pooling as the readout function in Equation (4), $\mathbf{z}_{\mathbf{k}'}^{\mathbf{0}} = \sum_{i \in C_{k'}} \mathbf{h}_i^{L_1}$, and multiplies the MHA output in Equation (7) with the number $k$ of patches, $\mathbf{q}_\ell' = k \cdot \mathrm{MHA}\left(\mathbf{q}_{\ell-1}, \mathbf{Z}_{L_2}, \mathbf{Z}_{L_2}\right)$. Let $\mathbf{g}_1$ and $\mathbf{g}_2$ be outputs computed from two graphs $G_1$ and $G_2$ by a PatchGT model. There exists a PatchGT such that $\mathbf{g}_1 \neq \mathbf{g}_2$ if $G_1$ and $G_2$ can be distinguished by the 1-WL algorithm. Furthermore, there are graph pairs $G_1 \neq G_2$ that cannot be distinguished by the 1-WL algorithm, but $\mathbf{g}_1 \neq \mathbf{g}_2$ from this PatchGT model.*

The first part of the conclusion is true because the patch aggregation, patch-level GNN, and the MHA pooling can all be bijective mapping. According to Corollary 6 of [36], the outputs of GIN layers have the same expressive power as the 1-WL algorithm. Such expressive power is maintained in the model output. However, when GIN layers on patches use extra structural information on patches, the

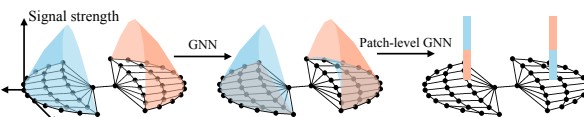

**Figure 2:** Pooling methods on a pair of graphs that cannot be distinguished by the 1-WL algorithm (nodes are colored by the 1-WL algorithm).

**Figure 3:** It is hard for a GNN to push signal from one graph cluster to the other, but a patch-level GNN can do so with patch representations.

model can distinguish graphs that cannot be distinguished by the 1-WL algorithm. We put the formal proof in Appendix A.4.

## 4.2 Permutation invariance

Our model depends on the patch structure formed by the clustering algorithm, which further depends on the spectral decomposition of the normalized Laplacian. Note that the spectral decomposition is not unique, but we show that the clustering result is not affected by sign variant and multiplicities associated with decomposition, and our model is still invariant to node permutations.

**Theorem 4.** *The network function of PatchGT is invariant to node permutations.*

## 4.3 Addressing information bottleneck with patch representations

Alon et al. [2] recently characterize the issue of information bottleneck in GNNs through empirical methods. Here we consider this issue on a special case when a graph consists of loosely-connected node clusters. Note that molecule graphs often have this property. Here we make the first attempt to characterize the *information bottleneck* through theoretical analysis. We further show that our PatchGT can partially address this issue.

For convenient analysis, we consider a regular graph with degree $\tau$. Suppose the node set $V$ of $G$ forms two clusters $S$ and $T$: $V = S \cup T$, $S \cap T = \emptyset$, and there are only $m$ edges between $S$ and $T$.

We consider the difficulty of passing signal from $S$ to $T$. Let $f^{\text{GNN}}(\cdot)$ denote the network function of a GNN of $L$ layers with ReLU activation $\sigma$ as in (2), and input $\mathbf{X} = (\mathbf{x}_i \in \mathbb{R}^d : i \in V) \in \mathbb{R}^{|V| \times d}$, which contains $d$-dimensional feature inputs to nodes in $G$. Let $f_i^{\text{GNN}}(\cdot)$ be the output at node $i$. We can ask this question: *if we perturb the input to nodes in $S$, how much impact we can observe at the output at nodes in $T$.* We need to avoid the case that the impact is amplified by scaling up network parameters. In real applications, scaling up network parameters also amplifies signals within $T$ itself, and the signal from $S$ still cannot be well received. Here we consider *relative impact*: the ratio between the impact on $T$ from $S$ over that from $T$ itself.

Let $\boldsymbol{\alpha} \in \mathbb{R}^{|V| \times d}$ be some perturbation on $S$ such that $\alpha_{ij} \leq \epsilon$ if $i \in S$ and $\alpha_{ij} = 0$ otherwise. Here $\epsilon$ is the scale of the perturbation. Similarly let $\boldsymbol{\beta} \in \mathbb{R}^{|V| \times d}$ be some perturbation on $T$: $\beta_{ij} \leq \epsilon$ if $i \in T$ and $\beta_{ij} = 0$ otherwise. Then the impacts on node representations $f_i^{\text{GNN}}$, $i \in T$ from $\boldsymbol{\alpha}$ and $\boldsymbol{\beta}$ are respectively

$$\delta_{S \to T} = \max_{\boldsymbol{\alpha}} \sum_{i \in T} \|f_i^{\text{GNN}}(\mathbf{X} + \boldsymbol{\alpha}) - f_i^{\text{GNN}}(\mathbf{X})\|_1 \tag{10}$$

$$\delta_{T \to T} = \max_{\boldsymbol{\beta}} \sum_{i \in T} \|f_i^{\text{GNN}}(\mathbf{X} + \boldsymbol{\beta}) - f_i^{\text{GNN}}(\mathbf{X})\|_1 \tag{11}$$

where the maximum is also over all possible learnable parameters $\|\mathbf{W}_1\|_{L_1 \to L_1}, \|\mathbf{W}_2\|_{L_1 \to L_1} \leq 1$ as in (2). Then we have the following proposition to bound the ratio $\delta_{S \to T}/\delta_{T \to T}$.

**Proposition 1.** *Given a $\tau$-regular graph $G$, a node subset $S$ with its complement $T$ such that there are only $m$ edges between $S$ and $T$, and a $L$-layer GNN, it holds that*

$$\frac{\delta_{S \to T}}{\delta_{T \to T}} \leq \frac{2mL}{|T|} \tag{12}$$

The proposition indicates that when there is a small graph cut between two clusters, then it forms an information bottleneck in a GNN – the network needs to use more layers to pass signal from one

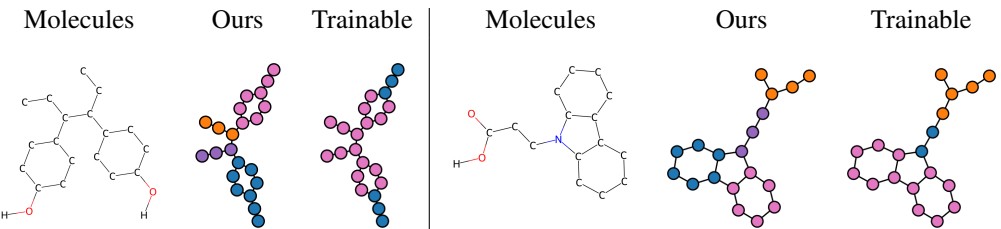

**Figure 4:** Segmentation results from spectral clustering and trainable clustering.

group to another. The bound is still conservative: if the signal is extracted in middle layers of the network, then passing the signal is even harder. The proposition is illustrated in Figure 3.

In our PatchGT model, communication can happen at the coarse graph and thus can partially address this issue. The coarse graph $\tilde{\mathbf{A}}$ consists of two nodes (we still denote them by $S, T$), and there is an edge between $S$ and $T$. From the output $f^{\text{GNN}}$, we construct the patch representations $(\mathbf{z}_S, \mathbf{z}_T) = (\frac{1}{|V|}\sum_{i \in S} f_i^{\text{GNN}}(\mathbf{X}), \frac{1}{|V|}\sum_{i \in T} f_i^{\text{GNN}}(\mathbf{X})) \in \mathbb{R}^{2 \times d}$. Then we apply a GNN layer to get node represents on the coarse graph $(g_S^{\text{GNN}}(\mathbf{X}), g_T^{\text{GNN}}(\mathbf{X})) \in \mathbb{R}^{2 \times d}$:

$$g_S^{\text{GNN}}(\mathbf{X}) = \sigma(\mathbf{z}_S \mathbf{W}_1^\top + \mathbf{z}_T \mathbf{W}_2^\top), \quad g_T^{\text{GNN}}(\mathbf{X}) = \sigma(\mathbf{z}_T \mathbf{W}_1^\top + \mathbf{z}_S \mathbf{W}_2^\top), \tag{13}$$

where $\mathbf{W}_1, \mathbf{W}_2 \in \mathbb{R}^{d \times d}$ are learnable parameters. We consider the impact of $\boldsymbol{\alpha}$ on our patch GT, let

$$\eta_{S \to T} = \max_{\boldsymbol{\alpha}} \|g_T^{\text{GNN}}(\mathbf{X} + \boldsymbol{\alpha}) - g_T^{\text{GNN}}(\mathbf{X})\|_1 \tag{14}$$

$$\eta_{T \to T} = \max_{\boldsymbol{\beta}} \|g_T^{\text{GNN}}(\mathbf{X} + \boldsymbol{\beta}) - g_T^{\text{GNN}}(\mathbf{X})\|_1, \tag{15}$$

Then we have the following proposition on the ratio $\eta_{S \to T}/\eta_{T \to T}$.

**Theorem 5.** *The ratio $\frac{\eta_{S \to T}}{\eta_{T \to T}}$ can be arbitrarily close to $1$ in a PatchGT model, under the assumption of regular graphs.*

This is because $S$ and $T$ are direct neighbors in the coarse graph, then $\alpha_S$ can directly impact $\mathbf{z}_S$, which can impact $g_T^{\text{GNN}}$ through messages passed by GNN layers or the attention mechanism of Transformer layers. The right part of fig. 3 shows that patch representation can include signals from the other node cluster.

## 4.4 Comparison for different Segmentation methods

In the previous researches, there exist many hierarchical pooling models [4, 12, 13, 19, 27, 39]. The most obvious difference from the proposed method is that the pooling/segmentation is trainable. Particularly, the pooling is from the node respresentations learned by GNNs. In the Theorem 1 and Theorem 2, we prove such trainable clustering methods will compute the same representations to the nodes if 1-WL algorithm can not differentiate them. This takes two serious problems for the graph segmentation: First, the nodes with the same representations will be assigned to the same cluster even if they are not connected to each other; Second, too many nodes could be assigned to one cluster to make sure that the nodes far away from each other are in the same cluster.

Here we compare the two segmentation results: one is from spectral clustering and another is from Memory-based graph networks[1] which is a typical trainable clustering method. In the first case, we find that nodes in the blue cluster from trainable clustering are not connected. If we adopt such patch representations by aggregating the disconnected nodes, it will definitely hurt the performance. This can also be applied to other hierarchical pooling methods such as Diffpool, Eigenpool, and MinCutpool.

In the second case, the spectral clustering methods segment the graph by minimum cuts. This is helpful to solve the information bottleneck between patches. However, the Memory-based graph networks cluster the two benzene rings together. It will be difficult for the model to detect the existence of these two benzene rings.

## 5 Empirical Study

In this section, we evaluate the effectiveness of PatchGT through experiments.

**Datasets.** We benchmark the performances of PatchGT on several commonly studied graph-level prediction datasets. The first four are from the Open Graph Benchmark (OGB) datasets [14] (ogbg-molhiv, ogbg-molbace, ogbg-molclintox, and ogbg-molsider). These tasks are predicting molecular attributes. The evaluation metric for these four datasets is ROC-AUC (%). The second group of six datasets are from the TU datasets [25], and they are DD, MUTAG, PROTEINS, PTC-MR, ENZYMES, and Mutagenicity. Each dataset contains one classification task for molecules. The evaluation metric is accuracy (%) over all six datasets. The statistics for the datasets is summarized in Appendix A.11.

### 5.1 Quantitative evaluation

**Table 1:** Results (%) on OGB datasets

|  | ogbg-molhiv | ogbg-molbace | ogbg-molclintox | ogbg-molsider |
|---|---|---|---|---|
| GCN +VN | 75.99 ±1.19 | 71.44 ± 4.01 | 88.55±2.09 | 59.84±1.54 |
| GIN + VN | 77.07±1.49 | 76.41±2.68 | 84.06±3.84 | 57.75 ±1.14 |
| Deep LRP | 77.19±1.40 | - | - | - |
| PNA | 79.05±1.32 | - | - | - |
| Nested GIN | 78.34±1.86 | 74.33±1.89 | 86.35±1.27 | 61.2±1.15 |
| GRAPHSNN +VN | 79.72±1.83 | - | - | - |
| Graphormer (pre-trained) | **80.51**±0.53 | - | - | - |
| PatchGT-GCN | 80.22±0.84 | 86.44±1.92 | **92.21** ±1.35 | 65.21 ± 0.87 |
| PatchGT-GIN | 79.99±1.21 | 84.08±2.03 | 86.75 ±1.04 | 64.90 ±0.92 |
| PatchGT-DeeperGCN | 78.13 ± 1.89 | **88.31**±1.87 | 89.02± 1.21 | **65.46**±1.03 |

**Table 2:** Results (%) on TU datasets

|  | DD | MUTAG | PROTEINS | PTC-MR | ENZYMES | Mutagenicity |
|---|---|---|---|---|---|---|
| GCN | 71.6±2.8 | 73.4±10.8 | 71.7±4.7 | 56.4±7.1 | 50.17 | - |
| GraphSAGE | 71.6±3.0 | 74.0±8.8 | 71.2±5.2 | 57.0±5.5 | 54.25 | - |
| GIN | 70.5±3.9 | 84.5±8.9 | 70.6±4.3 | 51.2±9.2 | 59.6 | - |
| GAT | 71.0±4.4 | 73.9±10.7 | 72.0±3.3 | 57.0±7.3 | 58.45 | - |
| DiffPool | 79.3±2.4 | - | 72.7±3.8 | - | 62.53 | 77.6±2.7 |
| MinCutPool | 80.8±2.3 | - | 76.5±2.6 | - | - | 79.9±2.1 |
| Nested GCN | 76.3±3.8 | 82.9±11.1 | 73.3±4.0 | 57.3±7.7 | 31.2±6.7 | - |
| Nested GIN | 77.8±3.9 | 87.9±8.2 | 73.9±5.1 | 54.1±7.7 | 29.0±8.0 | - |
| DiffPool-NOLP | 79.98 | - | 76.22 | - | 61.95 | - |
| SEG-BERT | - | 90.8 ±6.5 | 77.1±4.2 | - | - | - |
| U2GNN | 80.2±1.5 | 89.9±3.6 | 78.5±4.07 | - | - | - |
| EigenGCN | 78.6 | - | 76.6 | - | 64.5 | - |
| Graph U-Nets | 82.43 | - | 77.68 | - | - | - |
| PatchGT-GCN | **83.3**±3.1 | **94.7**±3.5 | 80.3±2.5 | **62.5**±4.1 | 73.3±3.3 | 78.3±2.2 |
| PatchGT-GIN | 79.6±3.3 | 89.4±3.2 | **79.5**±3.1 | 58.4±2.9 | **70.0**±3.5 | 80.4±1.4 |
| PatchGT-DeeperGCN | 76.1±2.8 | 89.4±3.7 | 77.5±3.4 | 60.0±2.6 | 56.6±3.1 | **80.6**±1.5 |

**Baselines.** In this section, we compare the performance of PatchGT against several baselines including GCN [17], GIN [36], as well as recent works Nested Graph Neural Networks [44] and GraphSNN [33]. To compare with learnable pooling methods, we also include DiffPool [39], MinCutPool [4] Graph U-Nets[12], and EigenGCN[23] as baselines for TU datasets. We also include the Graphormer model, but note that Graphormer needs a large-scale pre-training and cannot be easily applied to a wider range of datasets. We also compare our model with other transformer-based models such as U2GNN[26] and SEG-BERT[43].

**Settings.** We search model hyper-parameters such as the eigenvalue threshold, the learning rate, and the number of graph neural network layers on the validation set. Each OGB dataset has its own data split of training, validation, and test sets. We run ten fold cross-validation on each TU dataset. In each fold, one-tenth of the data is used as the test set, one-tenth is used as the validation set, and the rest is used as training. For the detailed search space, please refer to Appendix A.12.

**Figure 5:** Analysis of the key design for the proposed PatchGT. All results are based on PatchGT GCN. In the left figure, we show how changing the threshold for eigenvalues affects performance on the ogbg-molclintox and PROTEINS datasets; The middle figure shows the model performances with the removal of patch-GNN or Transformer (replaced by mean pool) on DD and ogbg-molhiv datasets; The right figure shows the effect of the different readout functions for patch representations.

**Figure 6:** Attention visualization of PatchGT on ogbg-molhiv molecules. The second and fourth figures show the attention weights of query tokens on the node patches for the corresponding molecules, which are in the first and third figures. The molecule in the first figure does not inhibit HIV virus, yet the molecule in the third figure does.

**Results.** Table 1 and Table 2 summarize the performance of PatchGT and other baselines on OGB datasets and TU datasets. We take values from the original papers and the OGB website; EXCEPT the performance values of Nested GIN on the last three OGB datasets – we obtain the three values by running Nested GIN. We also tried to run the contemporary method GRAPHSNN+VN on the other three OGB datasets, but we did not find the official implementation at the submission of this work.

From the results, we see that the proposed method gets good performances on almost all datasets and often outperforms competing methods with a large margin. On the ogbg-molhiv dataset, the performance of PatchGT with GCN is only slightly worse than Graphormer, but note that Graphormer needs large-scale pre-training, which limits its applications.

PatchGT with GCN outperforms three baselines on the other three OGB datasets. The improvements on these three OGB datasets are significant. PatchGT with GCN outperforms baselines on four out of six TU datasets. When it does not outperform all baselines, its performances are only slightly worse than the best performance. Similarly, two other configurations, PatchGT-GIN and PatchGT-DeeperGCN, also perform very well on these two datasets.

## 5.2 Ablation study

We perform ablation studies to check how different configurations of our model affect its performance. The results are shown in Figure 5.

**Effect of eigenvalue threshold.** The eigenvalue threshold $\gamma$ influences how many patches for a graph after the segmentation. Generally speaking, larger $\gamma$ introduces more patches and patches with smaller sizes. When $\gamma$ is large enough, the number of patches $k$ equals the number of nodes $|V|$ in the graph, and the Transformer actually works at the node level. When the $\gamma$ is 0, then the whole graph is treated as one patch, and the model is reduced to a GNN with pooling. The left figure shows that there is a sweet point (depending on the dataset) for the threshold, which means that using patches is a better choice than not using patches.

**Effect of GNN layer on the coarse graph and Transformer layers.** This ablation study removes either patch-level GNN layers or Transformer layers to check which part of the architecture is important for the model performance. From the middle plot in Figure 5, we see that both types of layers are useful, and Transformer layers are more useful. This is another piece of evidence that PatchGT can combine the strengths of different models.

**Comparison of readout functions.** We compare the performance of PatchGT model using different readout functions when aggregating node representations at each patch in Equation (4). In the right figure, we observe the remarkable influence of the readout function on the performance. Empirical studies indicate max-pooling is the optimal choice under most circumstances.

### 5.3 Understanding the attention

Besides improving learning performances, we are also interested in understanding how the attention mechanism helps the model identify the graph property. We train the PatchGT model on the ogbg-molhiv dataset and visualize the attention weights between query tokens and each patch. Interestingly, the attention only concentrates on some chemical motifs such as $Cl\,O_3$ and $CON_2$ but ignores other very common motifs such as benzene rings. It can be noticed that for the molecule in the first figure, the two benzene rings are connected to each other by -C-C-. However, the model does not pay any attention to this part. The two rings in the molecule of the second molecule are connected by -S-S-; differently, the model pays attention to this part this time. It indicates that Transformer can identify which motifs are informative and which motifs are common. Such property offers better model interpretability compared to the traditional global pooling. It not only makes accurate predictions but also provides some insight into why decisions are made. In the two examples shown above, we can start from motifs $SO_3$ and -S-S- to look for structures meaningful for the classification problem.

## 6    Conclusion and Limitations

In this work, we show that graph learning models benefit from modeling patches on graphs, particularly when it is combined with Transformer layers. We propose PatchGT, a new learning model that uses non-trainable clustering to get graph patches and learn graph representations based on patch representations. It combines the strengths of GNN layers and Transformer layers and we theoretically prove that it helps mitigate the bottleneck of graphs and limitations of trainable clustering. It shows superior performances on a list of graph learning tasks. Based on graph patches, Transformer layers also provides a good level of interpretability of model predictions.

However, the work tested our model mostly on chemical datasets. It is unclear whether the model still performs well when input graphs do not have clear cluster structures.

### Acknowledgements

Li-Ping Liu was supported by NSF1908617. Xu Han was supported by Tufts RA support. The research of J.H. is supported by NSF grant DMS-2054835. J.W and H.G would like to acknowledge the funds from National Science Foundation under Award Nos. CMMI-1934300 and OAC-2047127.

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

# A  Appendix

## A.1  Proof of Theorem 1

The proof that DiffPooling cannot distinguish graphs that are colored in the same way by the 1-WL algorithm.

*Proof.* The function form of a pooling layer in DiffPooling is

$$\mathbf{H}' = \mathbf{S}^\top \mathbf{A} \mathbf{S} \mathbf{S}^\top \mathbf{H}, \quad \mathbf{S} = \mathrm{gnn}_c(\mathbf{A}, \mathbf{X}), \quad \mathbf{H} = \mathrm{gnn}_r(\mathbf{A}, \mathbf{X}) \tag{16}$$

Here $\mathrm{gnn}_c(\cdot, \cdot)$ learns a cluster assignment $\mathbf{S}$ of all nodes in the graph, and $\mathrm{gnn}_r(\cdot, \cdot)$ learns node representations.

Note that $\mathrm{gnn}_r$ has at most the ability of 1-WL algorithm [36]. Two nodes must get the same representation when they have the same color in the 1-WL coloring result. We use an indicator matrix $\mathbf{C}$ to represent the 1-WL coloring of the graph, that is, the node $i$ is colored as $j$ if $C_{i,j} = 1$, then we can write

$$\mathbf{S} = \mathbf{C}\mathbf{B} \tag{17}$$

Here the $j$-th row of $\mathbf{B}$ denote the vector representation learned for color $j$.

If two graphs represented by $\mathbf{A}$ and $\mathbf{\Lambda}$ cannot be distinguished by the 1-WL algorithm, then they get the same coloring matrix $\mathbf{C}$ (subject to some node permutation that does not affect our analysis here). Now we show that:

$$\mathbf{C}^\top \mathbf{A} \mathbf{C} = \mathbf{C}^\top \mathbf{\Lambda} \mathbf{C} \tag{18}$$

Let's compare the two matrices on both sides of the equation at an arbitrary entry $(k, t)$. Let $\alpha_k$ and $\alpha_t$ represent nodes colored in $k$ and $t$, then the entry at $(k, t)$ is $\sum_{i \in \alpha_k} \sum_{j \in \alpha_t} A_{i,j}$, which is the count of edges that have one incident node colored in $k$ and the other incident node colored in $t$. Since the coloring is obtained by 1-WL algorithm, each node $i \in \alpha_k$ has exactly the same number of neighbors colored as $t$. The number of nodes in color $k$ and the number of neighbors in color $t$ are exactly the same for $\mathbf{\Lambda}$ because $\mathbf{\Lambda}$ receives the same coloring as $\mathbf{A}$. Therefore, $\sum_{i \in \alpha_k} \sum_{j \in \alpha_t} A_{i,j} = \sum_{i \in \alpha_k} \sum_{j \in \alpha_t} \Lambda_{i,j}$, and (18) holds.

At the same time, if two graphs cannot be distinguished by 1-WL, they have the same node representations $\mathbf{H}$, then they have the same $\mathbf{H}'$. □

## A.2  Proof of Theorem 2

We first prove a lemma.

**Lemma 1.** *Suppose two graphs represented by $\mathbf{A}$ and $\mathbf{\Lambda}$ obtain the same coloring from the 1-WL algorithm, then*

  i) *the resultant two graphs from removal of nodes in the same color still get the same coloring by the 1-WL algorithm; and*

  ii) *the two multigraphs represented by $\mathbf{A}^\ell$ and $\mathbf{\Lambda}^\ell$ still get the same coloring by the 1-WL algorithm.*

Here $\mathbf{A}^\ell$ and $\mathbf{\Lambda}^\ell$ are the $\ell$-th power of the two adjacency matrices, and they represent multigraphs that may have self-loops and parallel edges. The 1-WL algorithm is still valid over graphs with self-loops and multi-edges. A 1-WL style GNN defined in Section 3.1 or [12] is still bounded by the 1-WL algorithm on such multigraphs.

*Proof.* i) We first consider updating of 1-WL coloring when nodes in a color is removed. Suppose we have stable coloring of graphs represented by $\mathbf{A}$. Let $\alpha_t$ and $\alpha_r$ denote two groups of nodes in color $t$ and $r$ respectively. We also assume each node in $r$ has $t$ in its color set – if there are not such cases, then we can simply remove nodes in a color and obtain a stable 1-WL coloring.

Suppose we remove nodes in color $t$ from both graphs. Note that all nodes $\alpha_r$ have the same number of neighbors in color $t$. We update the color set of each $i \in \alpha_r$ by removing color $t$ from it. Then all nodes in $\alpha_r$ still get the same color. Therefore, removing the color $t$ from nodes in all relevant color groups gives at least a stable coloring, which, however, might not be the coarsest.

Then we merge some colors when nodes share the same color set. If a node in color $r$ has the same color set as a node in color $r'$, then we assign the same color to both nodes in colors $r$ and $r'$. We run merging steps until no nodes in different colors share the same color set, then the coloring is a stable coloring of the graph, and the resultant coloring of the graph can be viewed as the 1-WL coloring of the graph.

In the procedure above, the step of removing a color, and the steps of merging colors directly operate on nodes' color sets. Since nodes in $\mathbf{A}$ and nodes in $\mathbf{\Lambda}$ have the same color sets, therefore, they will have the same color sets after color updates.

The update procedure above purely runs on color relations between different colors. Since $\mathbf{A}$ and $\mathbf{\Lambda}$ have exactly the same color relations because they receive the same 1-WL coloring. Therefore, the update procedure above still gives the same stable coloring to $\mathbf{A}$ and $\mathbf{\Lambda}$.

ii) For the second part of the lemma, we first check the coloring of $\mathbf{A}^\ell$. We show that the coloring of $\mathbf{A}$ is a stable coloring of $\mathbf{A}^\ell$. Suppose each node $i$ has a color set $C_i$. In the graph $\mathbf{A}^\ell$, $i$'s $\ell$-th neighbors become direct neighbors of $i$. The color set of $i$ becomes

$$C_i \cup \left(\cup_{j_1 \in N(i)} C_j\right) \cup \ldots \cup \left(\cup_{j_1 \in N(i)} \ldots \cup_{j_\ell \in N(j_{\ell-1})} C_{j_\ell}\right) \tag{19}$$

We know that if two nodes $i$ and $i'$ have the same color if and only if their color sets are the same. By using the relation recursively, $i$ and $i'$ have the same color set in $\mathbf{A}^\ell$. Therefore, the stable coloring of $\mathbf{A}$ is also a stable coloring of $\mathbf{A}^\ell$. If necessary, we can also run the merging procedure above and eventually get 1-WL coloring of $\mathbf{A}^\ell$. With the same argument as above, the operations only run on color sets, therefore, $\mathbf{A}^\ell$ and $\mathbf{\Lambda}^\ell$ have the same coloring. $\square$

Now we are ready to prove the main theorem that the Graph U-Net variant cannot distinguish graphs colored in the same way by the 1-WL algorithm.

*Proof.* In the calculation of Graph U-Net-th, the indicator $\mathbf{b}$ for removing nodes is obtained by thresholding $\mathbf{v}$, which is computed by a 1-WL GNN. Therefore, nodes in the same color are always kept or removed all together in $\mathbf{b}$.

Suppose the inputs to a Graph U-Net layer are $(\mathbf{A}, \mathbf{X})$ and $(\mathbf{\Lambda}, \mathbf{X})$ respectively, and $\mathbf{A}$ and $\mathbf{\Lambda}$ cannot be distinguished by the 1-WL algorithm. The inputs to next layer are $(\mathbf{A}^\ell[\mathbf{b}, \mathbf{b}], \mathbf{X}[\mathbf{b}])$ and $(\mathbf{\Lambda}^\ell[\mathbf{b}, \mathbf{b}], \mathbf{X}[\mathbf{b}])$ respectively. By the lemma above, the 1-WL algorithm cannot distinguish $\mathbf{A}^\ell$ and $\mathbf{\Lambda}^\ell$, and it cannot be distinguish $\mathbf{A}^\ell[\mathbf{b}, \mathbf{b}]$ and $\mathbf{\Lambda}^\ell[\mathbf{b}, \mathbf{b}]$ either. Therefore, it still cannot distinguish the inputs $(\mathbf{A}^\ell[\mathbf{b}, \mathbf{b}], \mathbf{X}[\mathbf{b}])$ to the next layer.

By using the argument above recursively, the network cannot distinguish the graph at the final outputs if network inputs $(\mathbf{A}, \mathbf{X})$ and $(\mathbf{\Lambda}, \mathbf{X})$ cannot be distinguished by the 1-WL algorithm. $\square$

**Remark 1.** *For graphs with noise or low homophily ratios, the aforementioned issue may not be severe and long-distance aggregation is helpful.*

### A.3 Analysis for expressiveness of Graph U-Nets

In this section we use an example in Fig. 7 to understand how to maintain a graph's global structure with pooling operations. In a pooling step, DiffPool and MinCutPool will assign nodes in the same color to the same cluster and merge them as one node. Clearly it does not maintain the global structure of the graph and cannot distinguish the two graphs.

Graph U-Net always ranks nodes in one color above nodes of the other color. It is not always permutation invariant: for example, it may get different structures when it breaks tie to take two green nodes. In many cases, it cannot distinguish the two graphs: when it takes three nodes, either three green nodes or two blue and one green nodes, it cannot distinguish the two graphs. The Graph U-Net variant considered above always remove blue or green nodes, thus it cannot distinguish the two graphs. One important observation is Graph U-Net cannot preserve the global graph structure in its pooling steps. For example, when it removes three nodes, the structure left is vastly different from the original graph.

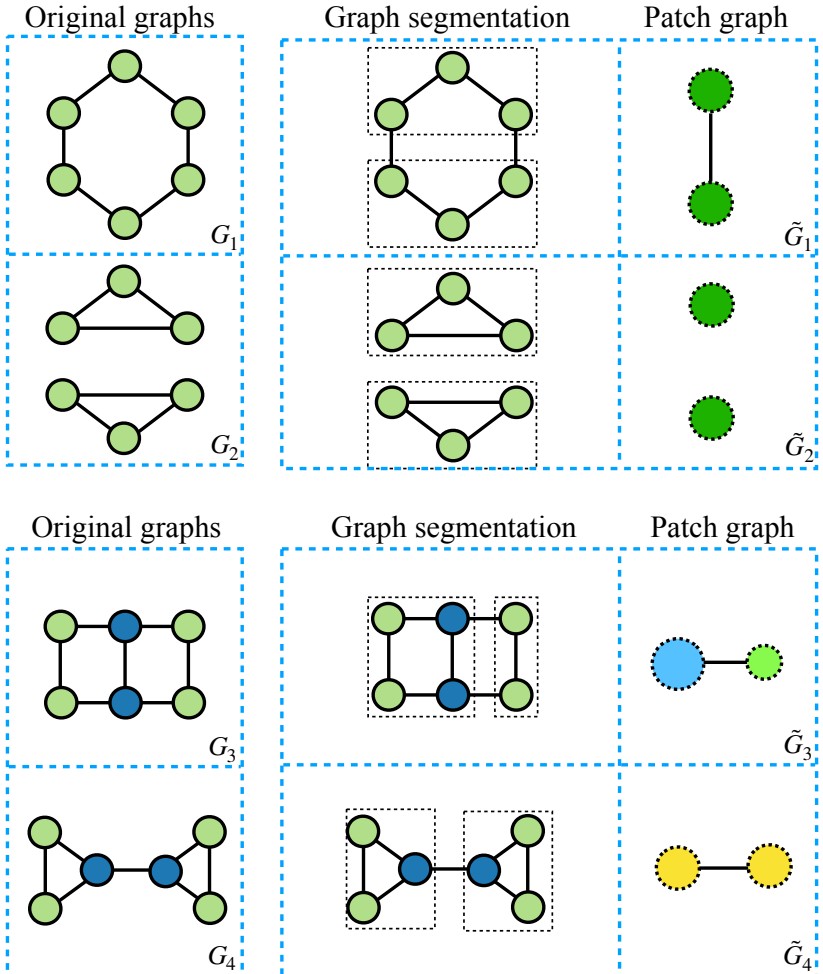

**Figure 7:** Two graphs that cannot be distinguished by the 1-WL algorithm. The colors illustrate the 1-WL coloring of graph nodes. In comparison, PatchGT can differentiate them through the patch-level graph.

### A.4 A proof showing that PatchGT is more expressive than the 1-WL algorithm

*Proof.* From the proof of GIN, we know that the two multi-sets $\{\mathbf{h}_i^{L_1} : i \in G_1\}$ and $\{\mathbf{h}_i^{L_1} : i \in G_2\}$ are already different if the two graphs can be distinguished by the $L_1$-round 1-WL algorithm.

Then we show that the rest of a learned network from Equation (4) to Equation (9) is a bijective operation. We first consider the patch aggregation by the sum-pooling is bijective. According to Corollary 6 of [36], and assuming the GIN layers are properly trained, then there is an inverse $\mathrm{inv}(\cdot)$ of sum-pooling such that $\{\mathbf{h}_i^{L_1} : i \in C_{k'}\} = \mathrm{inv}(\mathbf{z_0})$. Then the inverse of patch aggregation is:

$$\{\mathbf{h}_i^{L_1} : i \in G_1\} = \cup_{k'=1}^{k}\mathrm{inv}(\mathbf{z}_{k'}^0) \tag{20}$$

If the $L_2$ GNN layers on patches are also properly trained, then the mapping from $\mathbf{Z}_0$ to $\mathbf{Z}_{L_2}$ is also bijective. At the same time, we assume vectors in $\mathbf{Z}_{L_2}$ are properly transformed, which will be useful in the following MHA operation.

Finally, we consider MHA layers. We first analyze the case with only one layer with one attention head. Note that $\mathbf{q}_1 = k \cdot \mathrm{softmax}\left(\mathbf{q}_0^\top \mathbf{Z}_{L_2}/\sqrt{d}\right)^\top \mathbf{Z}_{L_2}$ with $d$ being the dimension of row vectors in $\mathbf{Z}_1$. Suppose PatchGT learns the query $q_0$ to be a zero vector, and the linear transformation in

Equation (9) is the identity operation, then $\mathbf{g}_1 = \mathbf{q}_1 = \mathbf{1}^\top \mathbf{Z}_{L_2}$, which is the summation of patch vectors $\mathbf{Z}_{L_2}$. Combining the last GIN layer, this summation is a bijective operation according to Corollary 6 of [36]. If there are multiple MHA layers, then we only need the MLP in Equation (8) to zero out the input, and the layer is equivalent to no operation. If there are multiple attention heads, the network can always take the first attention head. Therefore, a general case of MHA layers can also be a summation of input vectors.

Putting these steps together, there is an inverse mapping $\mathbf{g}_1$ to $\{\mathbf{h}_i^{L_1} : i \in G_1\}$ and mapping $\mathbf{g}_2$ to $\{\mathbf{h}_i^{L_1} : i \in G_2\}$. Then $\mathbf{g}_1$ and $\mathbf{g}_2$ must be different.

We further show that there are cases that cannot be distinguished by the 1-WL algorithm but can be distinguished by PatchGT. Consider two examples in Figure 7. The two original graphs $G_1$ and $G_2$, or $G_3$ and $G_4$, are non-isomorphic. However, both the 1-WL algorithm cannot differentiate them. In comparison, by segmenting these graphs into patches, PatchGT can discriminate $G_1$ from $G2$. After segmentation, the two patches from $G_1$ and the pacthes $G_2$ can be distinguished by the 1-WL algorithm and also PatchGT. Note that node degrees of $G_1$ patches are already different from node degrees of $G_2$ patches. It is the same for $G_3$ and $G_4$. These two examples indicate that the expressiveness of PatchGT is beyond 1-WL algorithm.

$\square$

## A.5 Proof of Theorem 4

We prove the theorem 4 through three lemmas below.

**Lemma 2.** *The patches split via $k$-means are invariant to column vectors in $\mathbf{U}$ from the spans of eigenvectors associated with the multiplicities of eigenvalues.*

$$\text{kmeans}(\mathbf{V}) = \text{kmeans}(\mathbf{VQ}) \tag{21}$$

*where $\mathbf{Q}$ is a standard block-diagonal rotation matrix.*

*Proof.* If we use $N_u$ eigenvectors for the graph patch splitting, corresponding to the first $N_u$ smallest eigenvalues, we can write them as $(\lambda_1, \mathbf{u}_1), ..., (\lambda_{N_u}, \mathbf{u}_{N_u})$. If we have multiplicities in these eigenvalues, we can rotate the eigenvectors by a block-diagonal rotation matrix $\mathbf{Q} \in \mathbb{R}^{N_u \times N_u}$ to obtain another set of eigenvectors,

$$\mathbf{U}' = [\mathbf{u}'_1, ..., \mathbf{u}'_k] = [\mathbf{u}_1, ..., \mathbf{u}_k]\mathbf{Q} = \mathbf{UQ} \tag{22}$$

where $\mathbf{u}_i, \mathbf{u}'_i \in \mathbb{R}^{|V| \times 1}$. If we perform $k$-means on the row vectors of $[(\mathbf{u}_1)_i, ..., (\mathbf{u}_k)_{N_u}]$, we can write the nodes' coordinates as

$$[\mathbf{x}_1; ...; \mathbf{x}_{|V|}] = [\mathbf{u}_1, ..., \mathbf{u}_{N_u}]. \tag{23}$$

Similarly, we can write down the new coordinates after rotation as

$$[\mathbf{x}'_1; ...; \mathbf{x}_{|V|}] = [\mathbf{u}'_1, ..., \mathbf{u}'_{N_u}]. \tag{24}$$

From the above three equations, it holds that

$$[\mathbf{x}'_1; ...; \mathbf{x}_{|\mathcal{V}'|}] = [\mathbf{x}_1; ...; \mathbf{x}_{|\mathcal{V}|}]\mathbf{Q}. \tag{25}$$

So for $i, j \in \{1, ..., |V|\}$, we have

$$\mathbf{x}'_i = \mathbf{x}_i\mathbf{Q} \quad \mathbf{x}'_j = \mathbf{x}_j\mathbf{Q}. \tag{26}$$

The relative distance of new coordinates can be calculated as

$$(\mathbf{x}'_i - \mathbf{x}'_j)(\mathbf{x}'_i - \mathbf{x}'_j)^\top = (\mathbf{x}_i\mathbf{Q} - \mathbf{x}_j\mathbf{Q})(\mathbf{x}_i\mathbf{Q} - \mathbf{x}_j\mathbf{Q})^\top = (\mathbf{x}_i - \mathbf{x}_j)\mathbf{QQ}^\top(\mathbf{x}_i - \mathbf{x}_j)^\top. \tag{27}$$

From the property of the rotational matrix, we have

$$\mathbf{I} = \mathbf{QQ}^\top. \tag{28}$$

So it holds that

$$(\mathbf{x}'_i - \mathbf{x}'_j)(\mathbf{x}'_i - \mathbf{x}'_j)^\top = (\mathbf{x}_i - \mathbf{x}_j)(\mathbf{x}_i - \mathbf{x}_j)^\top. \tag{29}$$

So for any two node pair, the relative distance is preserved, thus it will not affect the $k$-means results. $\square$

**Lemma 3.** *The patches split via $k$-means are invariant to column vectors in $\mathbf{U}$ with different signs.*

*Proof.* The sign invariance is a special case of rotation invariance by taking $\mathbf{Q}$ as a diagonal matrix with entry $(\mathbf{Q})_{ii} \in \{-1, 1\}$ ☐

**Lemma 4.** *The patches split via $k$-means are invariant to the permutations of nodes*

$$\text{kmeans}(\mathbf{U}) = \text{kmeans}(\mathbf{PU}) \tag{30}$$

*where $\mathbf{P}$ is a permutation matrix.*

*Proof.* We denote $\mathbf{I}_{|V|} = [1, ..., 1]^\top \in \mathbb{R}^{|V| \times 1}$ For a permutation matrix $\mathbf{P}$ of $\mathbf{A}$, we have the corresponding permutation matrix $P$ such that

$$\mathbf{A}' = \mathbf{P}^\top \mathbf{A} \mathbf{P} \tag{31}$$

where $\mathbf{A}$ and $\mathbf{A}'$ are adjacency matrices of $G$ and $G'$ respectively. And the for the degree matrix of $G$ and $G'$

$$\mathbf{D} = \text{diag}(\mathbf{A}' \mathbf{I}_{|V|}), \ \mathbf{D}' = \text{diag}(\mathbf{A}' \mathbf{I}_{|V|}) \tag{32}$$

Substitute equation 31 into equation 32

$$\mathbf{D}' = \text{diag}(\mathbf{P}^\top \mathbf{A} \mathbf{P} \mathbf{I}_{|V|}) = \text{diag}(\mathbf{P}^\top \mathbf{A} \mathbf{P}(\mathbf{P}^\top \mathbf{I}_{|V|} \mathbf{P})) \tag{33}$$

From the symmetry of the permutation matrix, it holds that

$$\mathbf{P}^{-1} = \mathbf{P}^\top \tag{34}$$

Combine the above three equations, we can get

$$\mathbf{D}' = \mathbf{P}^\top \text{diag}(\mathbf{A} \mathbf{I}_{|V|}) \mathbf{P} = \mathbf{P}^\top \mathbf{D} \mathbf{P} \tag{35}$$

So the permuted Laplacian matrix is

$$\begin{aligned}
\mathbf{L}' &= \mathbf{I} - \mathbf{D}'^{-0.5} \mathbf{A}' \mathbf{D}'^{-0.5} = \mathbf{P}^\top \mathbf{I} \mathbf{P} - \mathbf{P}^\top \mathbf{D}^{-0.5} \mathbf{P} \mathbf{P}^\top \mathbf{A} \mathbf{P} \mathbf{P}^\top \mathbf{D}^{-0.5} \mathbf{P} \\
&= \mathbf{P}^\top (\mathbf{I} - \mathbf{D}^{-0.5} \mathbf{A} \mathbf{D}^{-0.5}) \mathbf{P} = \mathbf{P}^\top \mathbf{L} \mathbf{P}
\end{aligned} \tag{36}$$

Substitute into the Laplacian eigen decomposition, we have the equation

$$\mathbf{L}' - \lambda \mathbf{I} = \mathbf{P}^\top \mathbf{L} \mathbf{P}^\top - \mathbf{P}^\top \lambda \mathbf{I} \mathbf{P} = \mathbf{P}^\top (\mathbf{L} - \lambda \mathbf{I}) \mathbf{P} \tag{37}$$

and its algebraic form

$$\det(\mathbf{L}' - \lambda \mathbf{I}) = \det(\mathbf{P}^\top) \det(\mathbf{L} - \lambda \mathbf{I}) \det(\mathbf{P}) = \det(\mathbf{L} - \lambda \mathbf{I}), \tag{38}$$

so the eigenvalues are remaining invariant.

Next we look at the eigenvector. For a eigenvector of $bL'$, $(\lambda, \mathbf{u}')$, we have

$$\mathbf{L}' \mathbf{u}' = \lambda \mathbf{u}' \tag{39}$$

Combine with equation 36, we can get

$$\mathbf{P}^\top \mathbf{L} \mathbf{P} \mathbf{u}' = \lambda \mathbf{u}' \iff \mathbf{L}(\mathbf{P} \mathbf{u}') = \lambda(\mathbf{P} \mathbf{u}') \tag{40}$$

So we have the relation of two corresponding eigenvectors as

$$\mathbf{u} = \mathbf{P} \mathbf{u}' \iff \mathbf{u}' = \mathbf{P}^\top \mathbf{u} \tag{41}$$

So we have the relation for the node coordinate

$$[\mathbf{x}_1'; ...; \mathbf{x}_{|V|}'] = \mathbf{P}^T [\mathbf{x}_1; ...; \mathbf{x}_{|V|}]. \tag{42}$$

Thus there is a bijective mapping $\mathcal{B} : n \to m$ such that $(\mathbf{P})_{n\mathcal{B}(n)} = 1$ and $\mathbf{x}_n = \mathbf{x}_{\mathcal{B}(n)}'$. Then for any node pair $(i, j)$, we can find $(i', j') = (\mathcal{B}(i), \mathcal{B}(j))$ such that

$$\mathbf{x}_i = \mathbf{x}_{i'}', \quad \mathbf{x}_j = \mathbf{x}_{j'}', \tag{43}$$

then it clearly holds that

$$(\mathbf{x}_i - \mathbf{x}_j)(\mathbf{x}_i - \mathbf{x}_j)^\top = (\mathbf{x}_{i'}' - \mathbf{x}_{j'}')(\mathbf{x}_{i'}' - \mathbf{x}_{j'}')^\top. \tag{44}$$

So for any two node pair, the relative distance is preserved, thus it will not affect the $k$-means results. ☐

## A.6 Multi-head attention

Transformer [32] has been proved successful in the NLP and CV fields. The design of multi-head attention (MHA) layer is based on attention mechanism with Query-Key-Value (QKV). Given the packed matrix representations of queries $\mathbf{Q}$, keys $\mathbf{K}$, and values $\mathbf{V}$, the scaled dot-product attention used by Transformer is given by:

$$\text{ATTENTION}(\mathbf{Q}, \mathbf{K}, \mathbf{V}) = \text{softmax}\left(\frac{\mathbf{Q}\mathbf{K}^T}{\sqrt{D_k}}\right)\mathbf{V}, \tag{45}$$

where $D_k$ represents the dimensions of queries and keys.

The multi-head attention applies $H$ heads of attention, allowing a model to attend to different types of information.

$$\text{MHA}(\mathbf{Q}, \mathbf{K}, \mathbf{V}) = \text{CONCAT}\left(\text{head}_1, \ldots, \text{head}_H\right)\mathbf{W}$$
$$\text{where} \quad \text{head}_i = \text{ATTENTION}\left(\mathbf{Q}\mathbf{W}_i^Q, \mathbf{K}\mathbf{W}_i^K, \mathbf{V}\mathbf{W}_i^V\right), i = 1, \ldots, H. \tag{46}$$

## A.7 Proof of proposition 1

Given a $L$ layer GNN with uniform hidden feature and initial feature $\mathbf{H}_0 = \mathbf{X}$, for $l = 0, ..., L$, the recurrent output of a GNN layer $\mathbf{H}_{l+1}$ follows

$$\mathbf{H}_{l+1} = \sigma(\mathbf{H}_l\mathbf{W}_{1l}^\top + \mathbf{A}\mathbf{H}_l\mathbf{W}_{2l}^\top) \tag{47}$$

where $\mathbf{H}_l \in \mathbb{R}^{|V|\times d}$, $\mathbf{W}_{1l}, \mathbf{W}_{2l} \in \mathbb{R}^{d\times d}$. And then we introduce another recurrent relationship to track the output change of each layers propagated from an initial perturbation $\boldsymbol{\epsilon}_0 \in \mathbb{R}^{|V|\times d}$ on $\mathbf{H}_0$,

$$\boldsymbol{\epsilon}_{l+1} = \sigma(\mathbf{H}_l\mathbf{W}_{1l}^\top + \mathbf{A}\mathbf{H}_l\mathbf{W}_{2l}^\top + \boldsymbol{\epsilon}_l\mathbf{W}_{1l}^\top + \mathbf{A}\boldsymbol{\epsilon}_l\mathbf{W}_{2l}^\top) - \sigma(\mathbf{H}_l\mathbf{W}_{1l}^\top + \mathbf{A}\mathbf{H}_l\mathbf{W}_{2l}^\top). \tag{48}$$

We denote $|\cdot|$ as an operator to replace a matrix's $(\cdot)$ elements with absolute values and we write $|\mathbf{J}| \leq |\mathbf{K}|$ if $|(\mathbf{J})_{ij}| \leq |(\mathbf{K})_{ij}|$. Let $\mathbf{I}_S \in \mathbb{R}^{|V|\times 1}$ is an indicator vector of $S$ such that $(\mathbf{I}_S)_i = 1$ if $i \in S$ else 0. We firstly prove a lemma below.

**Lemma 5.** *Given $\boldsymbol{\epsilon}_0 = \boldsymbol{\alpha}$, it holds that*

$$|\boldsymbol{\epsilon}_l| \leq a_l\mathbf{I}_S\mathbf{V}_l^\top + \mathbf{r}_l \in \mathbb{R}^{|V|\times d} \text{ or } |(\boldsymbol{\epsilon}_l)_{ij}| \leq a_l(\mathbf{V}_l)_j(\mathbf{I}_S)_i + (\mathbf{r}_l)_{ij} \tag{49}$$

*where $a_l = \epsilon(\tau+1)^l$, $\mathbf{V}_l \in \mathbb{R}_+^{d\times 1}$, $||\mathbf{V}_l||_1 \leq d$ and $||\mathbf{r}_l||_1 \leq 2r\epsilon m(l+1)(\tau+1)^l$.*

*Proof.* We prove by induction. For $l = 0$, we can take $a_0 = \epsilon$, $\mathbf{V}_0 = \mathbf{I}_d = \underbrace{[1, ..., 1]}_{d}$ and $\mathbf{r}_0 = \mathbf{0}$, then it holds

$$|\boldsymbol{\epsilon}_0| \leq a_0\mathbf{I}_S\mathbf{V}_0^\top + \mathbf{r}_0. \tag{50}$$

From the recurrent relation in equation 48, it holds that

$$\boldsymbol{\epsilon}_{l+1} = \sigma((\mathbf{H}_l + \boldsymbol{\epsilon}_l)\mathbf{W}_{1l}^\top + \mathbf{A}(\mathbf{H}_l + \boldsymbol{\epsilon}_l)\mathbf{W}_{2l}^\top) - \sigma(\mathbf{H}_l\mathbf{W}_{1l}^\top + \mathbf{A}\mathbf{H}_l\mathbf{W}_{2l}^\top). \tag{51}$$

From the Lipschitz continuity of $\sigma$, it holds that

$$|\boldsymbol{\epsilon}_{l+1}| \leq |\boldsymbol{\epsilon}_l\mathbf{W}_{1l}^\top + \mathbf{A}\boldsymbol{\epsilon}_l\mathbf{W}_{2l}^\top|. \tag{52}$$

From the triangle inequality, we have

$$|\boldsymbol{\epsilon}_{l+1}| \leq |\boldsymbol{\epsilon}_l||\mathbf{W}_{1l}^\top| + \mathbf{A}|\boldsymbol{\epsilon}_l||\mathbf{W}_{2l}^\top|. \tag{53}$$

From the assumption the statement holds at $l$th layer, we have

$$(*) \quad |\boldsymbol{\epsilon}_l| \leq a_l\mathbf{I}_S\mathbf{V}_l^\top + \mathbf{r}_l. \tag{54}$$

Substitute equation 54 into equation 53, we have,

$$|\boldsymbol{\epsilon}_{l+1}| \leq (a_l\mathbf{I}_S\mathbf{V}_l^\top + \mathbf{r}_l)|\mathbf{W}_{1l}^\top| + \mathbf{A}(a_l\mathbf{I}_S\mathbf{V}_l^\top + \mathbf{r}_l)|\mathbf{W}_{2l}^\top| \tag{55}$$

Expand the above equation,

$$|\boldsymbol{\epsilon}_{l+1}| \leq a_l(\mathbf{A}\mathbf{I}_S)(\mathbf{V}_l^\top|\mathbf{W}_{2l}^\top|) + \mathbf{A}\mathbf{r}_l|\mathbf{W}_{2l}^\top| + a_l\mathbf{I}_S\mathbf{V}_l^\top|\mathbf{W}_{1l}^\top| + \mathbf{r}_l|\mathbf{W}_{1l}^\top| \tag{56}$$

Using the property of undirected $\tau$-graph, it holds that

$$\mathbf{A}\mathbf{I}_S = \tau\mathbf{I}_S - \sum_{(i,j)\in E, i\in S, j\in T} (\mathbf{E}_i - \mathbf{E}_j) = \tau\mathbf{I}_S + \mathbf{B}_S, \tag{57}$$

where we denote

$$\mathbf{B}_S = -\sum_{(i,j)\in E, i\in S, j\in T} (\mathbf{E}_i - \mathbf{E}_j), \tag{58}$$

and $\mathbf{E}_i, \mathbf{E}_j \in \mathbb{R}^{|V|\times 1}$ are unit vectors with $i$th and $j$th entry equal to 1 respectively. Then it is trivial to show that

$$||\mathbf{B}_S||_1 \le 2m. \tag{59}$$

Substitute equation 57 into equation 56, we have

$$|\boldsymbol{\epsilon}_{l+1}| \le a_l\tau\mathbf{I}_S\mathbf{V}_l^\top|\mathbf{W}_{2l}^\top| + a_l\mathbf{B}_S\mathbf{V}_l^\top|\mathbf{W}_{2l}^\top| + \mathbf{A}\mathbf{r}_l|\mathbf{W}_{2l}^\top| + a_l\mathbf{I}_S\mathbf{V}_l^\top|\mathbf{W}_{1l}^\top| + \mathbf{r}_l|\mathbf{W}_{1l}^\top|. \tag{60}$$

Let

$$\begin{aligned}
a_{l+1} &= (1+\tau)a_l, \\
\mathbf{V}_{l+1}^\top &= \frac{\tau}{\tau+1}\mathbf{V}_l^\top|\mathbf{W}_{2l}^\top| + \frac{1}{\tau+1}\mathbf{V}_l^\top|\mathbf{W}_{1l}^T|, \\
\mathbf{r}_{l+1} &= a_l\mathbf{B}_S\mathbf{V}_l^\top|\mathbf{W}_{2l}^\top| + \mathbf{A}\mathbf{r}_l|\mathbf{W}_{2l}^\top| + \mathbf{r}_l|\mathbf{W}_{1l}^\top|,
\end{aligned} \tag{61}$$

then we rewrite equation 60 as

$$|\boldsymbol{\epsilon}_{l+1}| \le a_{l+1}\mathbf{I}_S\mathbf{V}_{l+1}^\top + \mathbf{r}_{l+1} \tag{62}$$

From the assumption that

$$||\mathbf{W}_{1l}||_1 \le 1, \quad ||\mathbf{W}_{2l}||_1 \le 1, \tag{63}$$

we have

$$||(|\mathbf{W}_{1l}|)||_1 = ||\mathbf{W}_{1l}||_1 \le 1, \quad ||(|\mathbf{W}_{2l}|)||_1 = ||\mathbf{W}_{2l}||_1 \le 1. \tag{64}$$

So substitute equation 64, equation 59 and equation 54 into equation 61,

$$\begin{aligned}
a_{l+1} &= (\tau+1)a_l \le \epsilon(\tau+1)^{l+1} \\
||\mathbf{V}_{l+1}^\top|| &\le \frac{\tau}{\tau+1}||\mathbf{V}_l^\top||_1 + \frac{1}{\tau+1}||\mathbf{V}_l^\top|| \le d
\end{aligned} \tag{65}$$

and

$$\begin{aligned}
||\mathbf{r}_{l+1}||_1 &\le a_l||\mathbf{B}_S||_1||\mathbf{V}_l^\top||_1 + ||\mathbf{A}||_1||\mathbf{r}_l||_1 + ||\mathbf{r}_l||_1 \\
&\le 2a_lmd + (\tau+1)||\mathbf{r}_l||_1 \le 2md\epsilon(\tau+1)^l + (\tau+1)||\mathbf{r}_l||_1 \\
&\le 2md\epsilon(\tau+1)^l + 2md\epsilon(l+1)(\tau+1)^{l+1} \\
&\le 2md\epsilon(\tau+1)^{l+1} + 2md\epsilon(l+1)(\tau+1)^{l+1} = 2md\epsilon(l+2)(\tau+1)^{l+1}.
\end{aligned} \tag{66}$$

This finishes the induction. $\qquad\square$

The above lemma gives

$$\max_{\substack{||\mathbf{W}_{1l}||_1 \\ ||\mathbf{W}_{2l}||_1 \\ \boldsymbol{\alpha}}} |\boldsymbol{\epsilon}_l| \le \epsilon(\tau+1)^l\mathbf{I}_S\mathbf{V}_l^\top + \mathbf{r}_l \tag{67}$$

where $||\mathbf{V}_l^\top|| \le d$ and $||\mathbf{r}_l||_1 \le 2d\epsilon m(l+1)(\tau+1)^l$. So when only looking at indices $\epsilon_{ij}$ with $i \in T$, the first term vanishes and it holds that

$$\max_{\substack{||\mathbf{W}_{1l}||_1 \\ ||\mathbf{W}_{2l}||_1 \\ \boldsymbol{\alpha}}} \sum_{i\in T} |\boldsymbol{\epsilon}_l|_{ij} \le 2d\epsilon m(l+1)(\tau+1)^l \tag{68}$$

For the denominator, we simply construct $\mathbf{W}_{1l} = \mathbf{W}_{2l}$ as both identity matrix and take $\epsilon_0 = \boldsymbol{\beta}$. Then it simply holds that

$$|\boldsymbol{\epsilon}_0| = (1+\tau)^0\epsilon\mathbf{I}_T\mathbf{I}_d^\top \tag{69}$$

where $\mathbf{I}_T$ is the indicator vector on set $T$. Assume it holds,

$$\boldsymbol{\epsilon}_l = (1+\tau)^l \epsilon \mathbf{I}_T \mathbf{I}_d^\top \tag{70}$$

then from the Lipschitz continuity (ReLU) of $\sigma$ and standard $\tau$-graph, it holds that

$$\boldsymbol{\epsilon}_{l+1} = \sigma((\mathbf{I}+\mathbf{A})(\mathbf{H}_l + \boldsymbol{\epsilon}_l)) - \sigma((\mathbf{I}+\mathbf{A})(\mathbf{H}_l)) = (1+d)^l \epsilon (\mathbf{I}+\mathbf{A}) \mathbf{I}_T \mathbf{I}_d^\top = (1+\tau)^{l+1} \epsilon \mathbf{I}_T \mathbf{I}_d^\top \tag{71}$$

So we can get

$$\sum_{i \in T} |(\boldsymbol{\epsilon}_l)_{ij}| = \epsilon(1+\tau)^l \sum_{i \in T} (\mathbf{I}_T \mathbf{I}_d^\top)_{ij} = (1+\tau)^l \epsilon |T| d \tag{72}$$

So that it holds that

$$\max_{\substack{||\mathbf{W}_{1l}||_1 \\ ||\mathbf{W}_{2l}||_1 \\ \boldsymbol{\beta}}} \sum_{i \in T} |\boldsymbol{\epsilon}_l|_{ij} \geq (1+\tau)^l \epsilon |T| d \tag{73}$$

Combine equation 68 and equation 73, and substitute the last layer number as $L-1$, we have

$$\frac{\max\limits_{\substack{||\mathbf{W}_{1l}||_1 \\ ||\mathbf{W}_{2l}||_1 \\ \boldsymbol{\alpha}}} \sum_{i \in T} |\boldsymbol{\epsilon}_l|_{ij}}{\max\limits_{\substack{||\mathbf{W}_{1l}||_1 \\ ||\mathbf{W}_{2l}||_1 \\ \boldsymbol{\beta}}} \sum_{i \in T} |\boldsymbol{\epsilon}_l|_{ij}} \leq \frac{2mL}{|T|}. \tag{74}$$

## A.8 Proof of Theorem 5

From the proof of proposition 1 in appendix A.7, by simply constructing $\mathbf{W}_{1l}, \mathbf{W}_{2l}$ in the node-level GNN as identity matrix, we have

$$\begin{aligned}
\sum_{i \in S} |(\boldsymbol{\epsilon}_S)_{ij}| &= (1+\tau)^L \epsilon |S| d \quad \text{if } \boldsymbol{\epsilon}_0 = \boldsymbol{\alpha}, \\
\sum_{i \in T} |(\boldsymbol{\epsilon}_S)_{ij}| &= (1+\tau)^L \epsilon |T| d \quad \text{if } \boldsymbol{\epsilon}_0 = \boldsymbol{\beta}.
\end{aligned} \tag{75}$$

Then from Lipschitz continuity (ReLU) we have

$$\begin{aligned}
\eta_{S \to T} &= g_T^{\text{GNN}}(\mathbf{X} + \boldsymbol{\alpha}) - g_T^{\text{GNN}}(\mathbf{X}) \\
&= \sigma(\mathbf{z}_T \mathbf{W}_1^\top + (\mathbf{z}_S + \frac{1}{|V|} \sum_{i \in S} (\boldsymbol{\epsilon}_S)_{ij}) \mathbf{W}_2) - \sigma(\mathbf{z}_T \mathbf{W}_1^\top + \mathbf{z}_S \mathbf{W}_2^\top) \\
&= (\frac{1}{|V|} \sum_{i \in S} (\boldsymbol{\epsilon}_S)_{ij}) \mathbf{W}_2^\top \quad \text{if } \boldsymbol{\epsilon}_0 = \boldsymbol{\alpha}
\end{aligned} \tag{76}$$

and

$$||\eta_{S \to T}||_1 = ||(\frac{1}{|V|} \sum_{i \in S} (\boldsymbol{\epsilon}_S)_{ij}) \mathbf{W}_2^\top||_1 = ||\mathbf{W}_2^\top||_1 (1+\tau)^L \epsilon \frac{|S|}{|V|} d \quad \text{if } \boldsymbol{\epsilon}_0 = \boldsymbol{\alpha} \tag{77}$$

Similarly, we can get

$$\eta_{T \to T} = (\frac{1}{|V|} \sum_{i \in T} (\boldsymbol{\epsilon}_T)_{ij}) \mathbf{W}_1^\top. \quad \text{if } \boldsymbol{\epsilon}_0 = \boldsymbol{\beta}, \tag{78}$$

and

$$||\eta_{T \to T}||_1 = ||\mathbf{W}_1^\top||_1 (1+\tau)^L \epsilon \frac{|T|}{|V|} d \quad \text{if } \boldsymbol{\epsilon}_0 = \boldsymbol{\beta}, \tag{79}$$

Then we can simply make $\frac{||\mathbf{W}_1||_1}{||\mathbf{W}_2||_1} = \frac{|S|}{|T|}$, so that the ratio is 1.

**Remark 2.** *The assumption of output norm unification can be achieved by standard normalization, such as batch and layer normalizations. Lipschitz continuity exists widely in the activation functions such as ReLU. And most molecules can be modeled as qusi standard graphs. These assumptions are fair assumptions in graph learning. Although it is difficult to universally obtain a precise and tight bound, the existence of such bounds is still helpful for GNN structure design.*

**Remark 3.** *The ratio may become informative if there are information bottlenecks within a cluster. We can mitigate the problem by having an appropriate, sufficient number of clusters. However, the number of clusters can not be too large, so there is a tradeoff between avoiding bottlenecks and computational cost.*

Here, we also introduce a heuristic example for possibly extending to a non-standard graph. Let subgraph $S$ be an cycle (2-graph) and subgraph $T$ be a clique ($n$-graph), approximately. And we assume $|S| = |T| = n$, and node values are all units with perturbation $\epsilon$. After one propagation of node level, each node in $S$ has the value $3(1 + \epsilon)$, each node in $T$ has the value $n(1 + \epsilon)$. Then at patch level, equations (75), (77), (79) are modified accordingly as $\eta_{S \to T} = 3\epsilon W_2$, $\eta_{T \to T} = n\epsilon W_1$, then $\frac{\eta_{S \to T}}{\eta_{T \to T}} = \frac{3}{n} \cdot \frac{W_2}{W_1}$, which indicates the unevenness may affect the performance. However, if at patch level $\frac{W_2}{W_1} \approx O(n)$ can be learned, we can still reach a sub-optimal balance. Actually, if $\frac{W_2}{W_1} > 1$ can be learned, it will help mitigate the bottleneck anyway.

### A.9   Graph segmentation

As a graph has an irregular structure and contains rich structural information, forming patches on a graph is not as straightforward as segmenting images. The previous works [9, 22] generally split an image in the euclidean space. However, graphs are segmented through spectral clustering based on its topology. Figure 8 shows the second eigenvector and patch segmentations based on the algorithm described in Section 3.2. It can be seen that the eigenvectors change along with the graph structures, and the graphs are splitted into several function groups. Such patches are useful for discriminating the property of the given molecule.

### A.10   More results

Table 7 provides the performance of PatchGT on ogbg-moltox21 and ogbg-moltoxcast.

**Table 3:** Results (%) on OGB datasets

|  | ogbg-moltox21 | ogbg-moltoxcast |
|---|---|---|
| GCN +VN | $75.51 \pm 0.86$ | $66.33 \pm 0.35$ |
| GIN + VN | $76.21 \pm 0.82$ | $66.18 \pm 0.68$ |
| GRAPHSNN +VN | $76.78 \pm 1.27$ | $67.68 \pm 0.92$ |
| PatchGT-GCN | $76.49 \pm 0.93$ | $66.58 \pm 0.47$ |
| PatchGT-GIN | $\mathbf{77.26} \pm 0.80$ | $\mathbf{67.95} \pm 0.55$ |

### A.11   Datasets

Table 4 contains the statistics for the six datasets from Open Graph Bechmark (OGB) [14], and Table 5 contains the statistics for the six datasets from TU datasets [25].

**Table 4:** Statistics of OGB datasets

| Name | #Graphs | #Nodes per graphs | #Edges per graph | #Tasks |
|---|---|---|---|---|
| molhiv | 41,127 | 25.5 | 27.5 | 1 |
| molbace | 1,513 | 34.1 | 36.9 | 1 |
| molclintox | 1,477 | 26.2 | 27.9 | 2 |
| molsider | 1,427 | 33.6 | 35.4 | 27 |
| ogbg-moltox21 | 7,831 | 18.6 | 19.3 | 12 |
| ogbg-moltoxcast | 8,576 | 18.8 | 19.3 | 617 |

### A.12   Hyper-parameters selection

We report the detailed hyper-parameter settings used for training PatchGT in Table 6. The search space for $\lambda$ is $\{0.1, 0.2, 0.4, 0.5, 0.8\}$.

**Figure 8:** Examples of eigenvectors, and graph patches for molecules.

**Table 5:** Statistics of TU datasets

| Name | #Graphs | #Nodes per graphs | #Edges per graph |
|---|---|---|---|
| DD | 1,178 | 284.3 | 715.7 |
| MUTAG | 188 | 17.9 | 19.8 |
| PROTEINS | 1,113 | 39.1 | 72.8 |
| PTC-MR | 344 | 14.3 | 14.7 |
| ENZYMES | 600 | 32.6 | 62.1 |
| Mutagenicity | 4,337 | 30.3 | 30.8 |

**Table 6:** Model Configurations and Hyper-parameters

|  | OGB | TU |
|---|---|---|
| # **GNN layers** | 5 | 4 |
| # **patch-GNN layers** | 2 | 2 |
| **Embedding Dropout** | 0.0 | 0.1 |
| **Hidden Dimension** $d$ | 512 | 256 |
| # **Attention Heads** | 16 | 4 |
| **Attention Dropout** | 0.1 | 0.1 |
| **Batch Size** | 512 | 256 |
| **Learning Rate** | 1e-4 | 1e-4 |
| **Max Epochs** | 150 | 50 |
| **eigenvalue threshold** $\lambda$ | $\{0.1, 0.2, 0.4, 0.5, 0.8\}$ | |

## A.13   Visualization of attention on nodes

Figure 9 shows more attention on graphs. We notice that some patches the model concentrates on are far away from each other. This can help address information bottleneck in the graph. Also, it provides more model interpretation.

## A.14   Analysis of the computational complexity

We compare our computational complexity with the node-level Transformer, Graphormer [38]. The comptutational complexity for both framework can be classified into two parts. The first part is extracting graph structure infromation. For PatchGT, the complexity is $O(|V|^3)$ for calculating the eigenvectors and perform kmeans for $k$ patches. For Graphormer, the complexity is $O(|V|^4)$ due to node pairwise shortest path computation.

**Remark 4.** *The software and algorithms of eigen-decomposition are being widely developed in many disciplines [10]. The complexity can be reduced to $O(|V|^2)$ if a partial query and approximation of eigenvectors and eigenvalues are allowed [8, 29]. And spectral clustering does not require all eigenvectors with exact values. However, we admit that for graphs with eigenvalues that are too close to each other, the complexity of computing the eigenvectors takes $O(N^3)$.*

The second part is neural network computation. For PatchGT, the complexity is $O(|E|)$ for GNN if the adjacency matrix is sparse and $O(k^2)$ for Transformer. And for Graphormer, the complexity of Transformer is $O(|V|^2)$. It shoud be noticed that for a large graph, $k << |V|$. Overall, the complexity of patch-level transformer is significantly less than that of applying transformer directly on the node level.

For other hierarchical pooling methods, they also need $O(L|E|)$ to learn the segmentation ($L$ is the number of layers used in GNN), which is comparable to spectral clustering. And spectral clustering is easier for parallel computation. Specifically, for a $N_{\text{pool}}$-level hierarchical pooling, it needs $O(\sum_{i=1}^{N_{\text{pool}}} L_i |E_i|)$ to learn the segmentation and $O(\sum_{i=1}^{N_{\text{pool}}} |V_i| d_i k_i)$ to perform the segmentation. When training epoch number becomes a large number, the extra accumulated cost is non-trivial. Our segmentation cost does not scale with the training iterations.

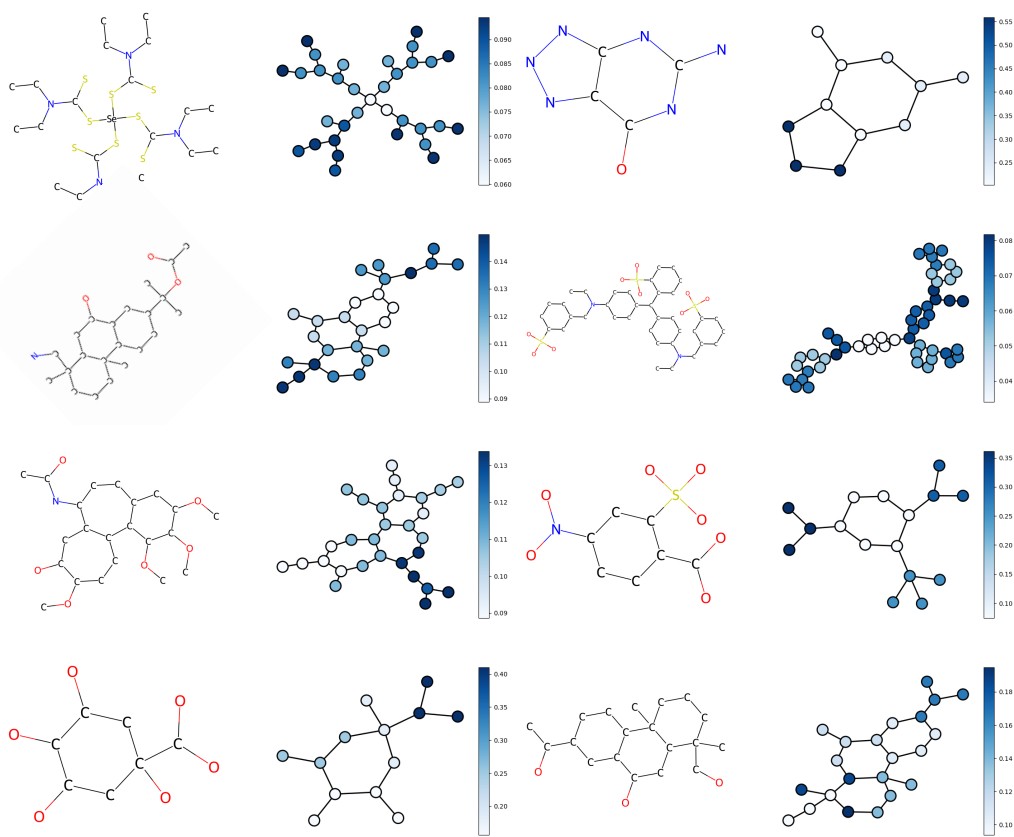

**Figure 9:** Attention visualization of PatchGT on ogbg-molhiv molecules.

## A.15 Frequencies of motifs

There are two classes in ogbg-molhiv, and we record the frequencies of motifs PatchGT pay attention to. There are an apparent difference between the two classes. It indicates the model has a better interpretability.

## A.16 Ablation study for patch level GNN

In PatchGT, we apply patch level GNN to the entire graph. We can also apply it to each patch so that there would not be any connection between subgraphs. Here we test the difference of these two designs.

**Table 7:** Results (%) on ogbg-molhiv

|  | single GNN | multiple GNNs |
|---|---|---|
| PatchGT-GCN | 80.22 ±0.0.84 | 79.13 ±0.47 |
| PatchGT-GIN | 79.99 ± 1.21 | 78.96 ±0.55 |

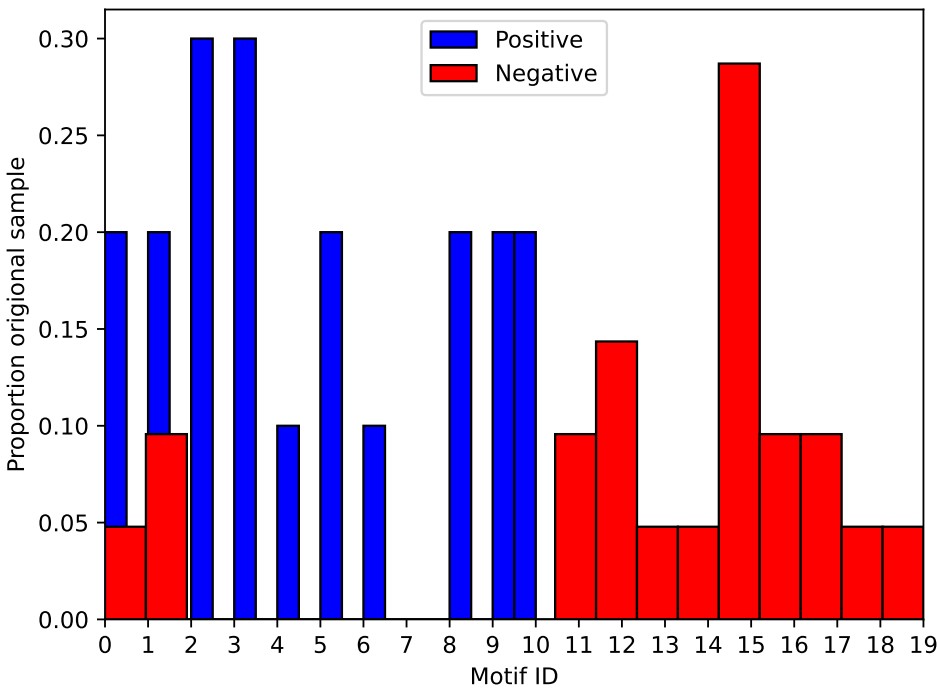

**Figure 10:** Frequency of motifs PatchGT pay attention to in two classes.

