# OpenReview forum: "PatchGT: Transformer over Non-trainable Clusters for Learning Graph Representations"
_logconference.io/LOG/2022/Conference — LoG 2022 Poster_

### Official Review · Reviewer_M7Tu · 2022-10-10

**Overall Score:** 3
**Confidence:** 4

**Review:**

Description:
The paper proposes to use transformers at a patch level rather than node level for the graph classification task. The patches are obtained using spectral clustering and the patch representations are computed using a GNN. The authors show that the proposed method is able to aggregate information over bottlenecks better than dominant message passing mechanisms. Experiments on the graph classification datasets show that the proposed method is able to perform on par or better than the baselines.

Strengths:
1) The paper attempts a theoretical support to the motivation, introducing a new metric to measure "information bottleneck"
2) The results on benchmarks are comparable to or better than baselines(mostly GNN).

Weaknesses:
1) There is no comparison to other GT models(SAN, GT etc.) except for Graphormer(and that too on just one dataset). GraphTrans(https://proceedings.neurips.cc/paper/2021/file/6e67691b60ed3e4a55935261314dd534-Paper.pdf) is another work that is related to the current work. GraphTrans uses a transformer for aggregating the features learnt by a GNN. These should be compared with as baselines at least on the smaller datasets.
2) The related work could be better. Please see relevant papers and explain what they do in transformers.
3) The authors mention that partitioning using GNNs is computationally costly. But the worst case complexity of computing the eigen decomposition(that the authors use) of the full graph laplacian takes O(N^3) time. So the method would only work effectively for small graphs. The approximations that the authors mention in the appendix are only under certain conditions that the laplacian does not have eigenvalues that are too close to each other etc.
4) Since the method compresses the graph, it can only be used for graph classification and can’t be used for applications such as node classification, link prediction etc.
5) Regarding the theory: The authors propose a metric to measure information bottleneck between the clusters. It is defined as the ratio of the max information difference of the two clusters. One issue with such a formulation is that if there are bottlenecks in the two clusters they will just be canceled out and the metric will be close to 1 which is not very desirable or informative of the bottlenecks within the clusters. Leaving this arguable concern, there seems to be a flaw in theorem 4. Consider a graph of 2n nodes with n nodes forming a cycle(S), n nodes forming a clique(T) and the two subgraphs connected by a single edge. Let each node have a signal value of 1. Now in order to maximize the norm of the difference between the information in the concerned cluster a perturbation +\epsilon is added to each node in S(or T). As in the proofs consider the weight matrices to be identity. After propagation the signal at each node of S becomes 3(1+\epsilon) and for T it takes value n(1+\epsilon). This is also the average value of the signal over the nodes for each cluster. Computing the terms \eta_{S->T} and \eta_{T->T} as defined and taking the ratio we get \frac{3}{n}, which cannot be arbitrarily close to 1 as the nodes increase. Thus we have a counter example. The issue may lie in the proof around eq 75 where linearity of the activation function is assumed and terms canceled. The statement that the ratio can get arbitrarily close to 1 is too broad and may hold under further assumptions that the clusters are increased etc. Diving deeper into these conditions may be useful for say a practitioner to decide the number of clusters based on the desired “information bottleneck” ratio as a tradeoff to the computational cost. The proof should be corrected for handling these cases.

Comments:
1) While the paper mentions the limitations of 1-WL with respect to clustering, the authors could also show the expressivity of the overall mode with respect to the WL test.
2) It seems the method does not make use of a position encoding(PE) for the transformer. Would it help enhance the results if a PE of the compressed graph were induced in the transformer?

Typos:
1) Note that patch representations Z_{L_2} are carried through without updated -> Note that patch representations Z_{L_2} are carried through without being updated
2) that PatchGT can distinguish but the 1-WL algorithm cannot in ?.
3) The coarse graph ˜A 210 consists of two notes -> The coarse graph ˜A 210 consists of two nodes
4) If we adopt such patch representations by aggregate the disconnected nodes, will definitely hurt the performance -> If we adopt such patch representations by aggregating the disconnected nodes, it will definitely hurt the performance
5) This is also can be applied to -> This can also be applied to



Missing Citations:
[1] Paras Jain, Zhanghao Wu, Matthew A. Wright, Azalia Mirhoseini, Joseph E. Gonzalez, & Ion Stoica (2021). Representing Long-Range Context for Graph Neural Networks with Global Attention. In Advances in Neural Information Processing Systems.



Overall the proposed work looks like a promising extension of the research on vision transformers(patchwise convolutions followed by transformers). However there is some more work to be done in terms of the evaluation the primary being comparison with the right baselines. This will help identify if the proposed method is beneficial or even needed for graphs over existing graph learning methods. Also the theory needs to be corrected/improved to maybe incorporate a criteria for selecting the number of clusters/eigenvalue threshold. Thus I would vote for a reject to allow the paper to be improved further.

---

### Official Review · Reviewer_TrWt · 2022-10-14

**Overall Score:** 6
**Confidence:** 5

**Review:**

**Contributions**

This paper studies the topic of hierarchical and global pooling in graph neural networks.

The contributions of the paper are:

1. A GNN architecture that combines 1) a standard GNN backbone with a non-trainable pooling step to coarsen graphs, and 2) a global readout based on cross-attention.
2. Several theorems related to the expressive power of GNNs with pooling, showing for the first time why (some) architectures with pooling are provably more powerful than architectures without.
3. Experiments on popular graph classification benchmarks where the proposed model achieves state-of-the-art performance, also supported by an ablation study of the main components of the architecture.

**Strengths**

1. The presentation is clear, although the manuscript could benefit from additional proof-reading to improve the syntax/grammar.
2. All design choices for the architecture are supported by experimental analysis.
3. This work is among the first to perform a sound theoretical analysis of the effect of pooling in graph neural networks and provides answers to multiple open questions on the topic. Specifically, it highlights some limitations of some state-of-the-art pooling methods and points at a possible solution (using methods with a "global" view of the graphs to overcome the limited expressive power of message-passing).
4. The proposed architecture achieves significantly better performance on typical benchmarks, including some OGB datasets.

**Weaknesses**

1. I think that the main narrative of the paper should be rethought. The paper is not about graph transformers (which typically focus on node-to-node self-attention), nor is it related to patch-based vision transformers (which would have patch-to-patch self-attention).
	Instead, the paper presents an analysis of the benefits of including non-trainable pooling (specifically, a pooling method that relies on global spectral information) in GNNs, with good arguments supported by the theory.
	Additionally, the paper studies how global readouts can be effectively implemented through a novel cross-attention mechanism that uses a fully learnable query vector, which is shown to provide significant benefits in the ablation study.
	For these reasons, I don't see a reason to frame this method as a "patch graph transformer", since the contributions of the paper are interesting enough even without this label.
2. The comment on the computational cost for trainable methods in Section 2 is not really well supported. All trainable pooling baselines mentioned in the paper have a cost of $O(n^2)$, which is the same cost of running spectral clustering on new graphs at inference time ($O(kn^2)$). Have the authors experienced concretely slower training times with trainable baselines?
3. At least one other graph readout based on attention has been proposed in

	> Li, Yujia, et al. "Gated graph sequence neural networks." _arXiv preprint arXiv:1511.05493_ (2015).

	I suggest the authors do a brief comparison of how the two approaches differ (possibly, even an experimental comparison).

**Recommendation**

The paper is interesting, but the main motivations and ideas behind the work are presented in a confusing/misleading way.
For this reason, I have recommended only a weak acceptance.

I suggest the authors re-frame the paper not as a work on "patch graph transformers", but as a work that makes several interesting observations on graph pooling.

**Additional feedback**

- Graphormer outperforms the proposed method on the only benchmark for which there are results. The authors should report Graphormer's results also on the other OGB benchmarks to ensure that their claim to SOTA holds.
- The claim on lines 118-119 that applying a GNN only within clusters gives similar performance should be supported by numbers/experimental results.
- Typos:
	- Line 87: denotes -> denote
	- Line 174: cannot in ... ?
	- Line 210: notes -> nodes

---

### Official Review · Reviewer_4TVh · 2022-10-21

**Overall Score:** 8
**Confidence:** 3

**Review:**

*Summary:* The paper proposes a graph transformer-based model for graph prediction tasks by taking inspiration from vision transformer models. In a first step, the presented approach clusters each graph using spectral clustering, resulting in patches of nodes. Patch representations are computed using a GCN both on the original graph and on the induced graph of patches. Those patches are used as inputs into the transformer architecture. The authors present theoretical results showing the benefit of using spectral clustering over GNN-based clustering methods with regards to the information bottleneck in GNNs. Empirical results on a range of benchmark data sets show strong performance compared to a number of strong baselines.

*Pros:*
* The paper is well written and easy to follow with a linear flow of information.
* The data sets are well chosen and extensive.
* The theoretical analysis is mostly informative and not too cluttered.

*Cons:*
* It is  unfortunate that there are a large number of gaps in the results tables in particular for methods which show competitive performance with the proposed method. This makes the results less convincing.

*Suggestions:*
* Section 5.3 is on somewhat weak footing. Interpretability is a strong claim and it would be nice to see this backed up by empirical evidence. One possibility could be to look at motifs with high attention weights and compare how often they occur in each class. If the model truly identifies “which motifs are informative and which motifs are common” (l. 303), then there should be clear differences in the frequencies of motifs across classes.

*Minor comments:*
* Typo in word “clusters” in line 74
* Erroneous capitalisation of “We” in line 14

Overall the paper is nicely written and a solid contribution to graph neural network literature. The proposed model and accompanying theoretical analysis should be interesting to the scientific community, hence I recommend acceptance.

---

### Official Review · Reviewer_3mXe · 2022-10-22

**Overall Score:** 6
**Confidence:** 4

**Review:**

Hello authors, thank you for your submission to LoG'22. I pen down my thoughts and considerations here:

**SUMMARY**

The authors introduce PatchGT, a graph transformer that segments a graph via spectral clustering to form “patches”, feeds these patches into a GNN for local patch-level representation, and finally a Transformer for global representations. The paper provides empirical and theoretical justifications on why former hierarchical method are lacking, and proves expressiveness beyond 1-WL GNNs. Another benefit is the easier interpretability of these PatchGT models.

**VERDICT**

I vote to (weak) reject this paper on a comparative basis with the existing literature. Taking a graph, converting it into patches (ie, a form of subgraphs), then processing these subgraphs separately (for local representations) and together (for global representations) has been done before. This paper presents an existing way (spectral clustering) to break up a graph and process it using a GNN and a Transformer.

**MERITS**

- Strong theoretical justifications on why previous hierarchical method are not the best when learning mid-level features from a graph.
- Strong theoretical justifications on > 1-WL expressiveness, while some existing Graph Transformers are as expressive as 1-WL.
- Reproducible results based on a quick run of some of the code, which is appealing in the sea of non-reproducible projects being introduced to the community nowadays.

**CONCERNS, IMPROVEMENTS, SUGGESTIONS**

1. The idea sounds similar to existing work that came out early in 2021 – Equivariant Subgraph Aggregation Networks (ESAN; Bevilacqua et al., 2021). ESAN randomly or systematically breaks a graph into subgraphs (which mimic the patches in PatchGT), and apply a H-equivariant layer (which is akin to your GNN layer). Your work seems close to ESAN except you have an additional Transformer on top to process the local patches (subgraphs). It’s interesting because ESAN is 3-WL expressiveness. Perhaps, you could convince me by showing that PatchGT outperforms ESAN on some existing benchmarks like ZINC?
2. Some of the proteins datasets considered for this task seem rather simplistic and erroneous. Do consider running your models on datasets from the “Benchmarking Graph Neural Networks” paper (Appendix F, Dwivedi et al., 2020) – it contains a diverse set of datasets that are popular and of high-standard. The MUTAG, PROTEINS, and D&D benchmarks are not representative of the complexities that exist within graph representation learning – MLPs do well on them, different random seeds affect performance, and the first one might be considered small.
3. Can I know how you think PatchGT will perform on heterophillic graphs that entails a large number of inter-class edges and fewer intra-class edges? Since you rely on spectral clustering, this method has issues when the density of inter-class edges is relatively large. Consider checking out this paper "Graph Attention Retrospective" (Fountoulakis et al., 2022). They show that models like GAT do not do well in heterophillic settings. I'd say models like GIN or GCN face similar issues which are worth investigating to see if PatchGT's GNN is actually robust to such situations.
4. In your conclusion (second line), do you mean PatchGT instead of GraphGT? Minor typo!

Thank you again for your contribution, it was a joy to read and definitely showcases how useful a Transformer is at extracting high-level features from graphs.

Best wishes and good luck!

---

### Meta-Review · Area_Chair_kAoq · 2022-11-17

**Confidence:** 4
**Recommendation:** Accept

**Meta Review:**

The reviewers agree on the paper presenting an intuitive and clear overview for graph transformer-based graph prediction tasks. Based on the reviews the contributions of the paper are now effectively presented, additional experiments are given and textual improvements are incorporated. The paper is overall to be considered sound. Although it contains some novel contributions which in itself may be considered incremental, its value may lie primarily in the theoretical and empirical analysis. Considering the paper is sound, clearly written and the experiments are appropriate, and moreover the paper is reproducible with easy to run code, I recommend accept.

---

### Decision · Program_Chairs · 2022-11-23

Accept (Poster)